# Geometric models for robust encoding of dynamical information into embryonic patterns

**Laurent Jutras-Dubé[1], Ezzat El-Sherif[2]\*, Paul François[1]\***

[1]Department of Physics, McGill University, Montreal, Canada; [2]Division of Developmental Biology, Department of Biology, Friedrich-Alexander-Universität Erlangen-Nürnberg, Erlangen, Germany

**Abstract** During development, cells gradually assume specialized fates via changes of transcriptional dynamics, sometimes even within the same developmental stage. For anterior-posterior (AP) patterning in metazoans, it has been suggested that the gradual transition from a dynamic genetic regime to a static one is encoded by different transcriptional modules. In that case, the static regime has an essential role in pattern formation in addition to its maintenance function. In this work, we introduce a geometric approach to study such transition. We exhibit two types of genetic regime transitions arising through local or global bifurcations, respectively. We find that the global bifurcation type is more generic, more robust, and better preserves dynamical information. This could parsimoniously explain common features of metazoan segmentation, such as changes of periods leading to waves of gene expressions, 'speed/frequency-gradient' dynamics, and changes of wave patterns. Geometric approaches appear as possible alternatives to gene regulatory networks to understand development.

**\*For correspondence:**
ezzat.el-sherif@fau.de (EE-S);
paul.francois2@mcgill.ca (PF)

**Competing interests:** The authors declare that no competing interests exist.

## Introduction

Development from one zygote to a viable animal is a complex process (*Wolpert et al., 2006*), comprising multiple dynamical sub-processes, including cell movements, tissue morphogenesis, dynamical gene expressions, and cellular differentiations. Eventually, cell identities are fixed by various mechanisms, such as multistable gene regulatory networks and epigenetic markers. Little is known about how this transition from a dynamic/initiation phase to a static/maintenance one is mediated. Are there general characteristics that should be matched between dynamic and static phases to mediate a robust transition?

In dynamical systems theory, a transition between different regimes is called a 'bifurcation', defined as a qualitative change in the dynamics of a system driven by a so-called 'control parameter' (*Strogatz, 2015*). Bifurcations are of many types but can be systematically classified. For instance, generic families of potentials driving the dynamics have been identified as different 'catastrophes' (*Poston and Stewart, 2012*). While such mathematical descriptions are highly technical, they are reminiscent of the theory of epigenetic landscapes pushed forward by *Waddington, 1957*. It is thus natural to ask if such classifications can be done for development. Could dynamical systems theory help us in this pursuit, and in studying development in general? Here, the main issue is to find a way to frame the problem to derive general results.

In recent years, numerous experimental studies have revealed that quantitative changes of gene expressions during development often followed standard predictions from dynamical systems theory (*Huang et al., 2007*). The Waddington landscape's analogy (*Jaeger and Monk, 2014*) has led to many insights in cell differentiation (*Graf and Enver, 2009*), and recent data on cell reprogramming quantitatively validated the associated 'landscape picture' (*Pusuluri et al., 2018*). Geometric models

of development have been developed in particular cases, precisely predicting the general phenotypes of wildtype and mutants (e.g. the development of *Caenorhabditis elegans* vulva [*Corson and Siggia, 2012*] and *Drosophila* bristle patterns [*Corson et al., 2017*]).

The Clock-and-Wavefront model (*Cooke and Zeeman, 1976*), accounting for the observed dynamical somite (vertebrae precursors) formation, was inspired by catastrophe theory. The model predicted that a retracting wavefront translates the periodic expression of a genetic clock into a spatial pattern via 'catastrophic' transitions demarcating the positions of the somites (*Figure 1A*). Identification of the predicted clock in 1997 (*Palmeirim et al., 1997*) has since led to many subsequent theoretical and experimental works, including observation of similar clocks in multiple arthropods (*El-Sherif et al., 2012*; *Sarrazin et al., 2012*). *Cooke and Zeeman, 1976* originally assumed that the clock is an external process, blind to the subsequent segmentation process it directs. However, it has been very clear from the early experiments in *Palmeirim et al., 1997* that cellular oscillators increase their period prior to segmentation, leading to traveling waves of various signalling pathways such as Notch (*Giudicelli et al., 2007*; *Morelli et al., 2009*; *Figure 1A*). Importantly, Notch waves eventually stabilize into a pattern of *delta* ligand stripes (*Giudicelli and Lewis, 2004*; *Jiang et al., 2000*), with a functional continuity between the dynamic and the static regime. Indeed, it has been shown that the dynamical phase of the clock is encoded into static rostro-caudal identities (*Oginuma et al., 2010*). This suggests that the observed oscillation is not a simple external pacemaker for segment formation: rather, clocks, associated waves and the resulting stripes combine into an emergent process leading to proper fate encoding. Segmentation thus possibly appears as the canonical example of transition from a dynamical gene expression regime to a static functional one.

Two broad scenarios have been proposed to model this process (see *Figure 1*) . In the first scenario, the period of the individual oscillators is diverging to infinity as they become more anterior (or similarly, the frequency of the clock is approaching zero), automatically giving rise to a fixed pattern (*Figure 1B–F*). This model corresponds to Julian Lewis' model for *c-hairy one* expression pattern in somitogenesis (appendix of [*Palmeirim et al., 1997*]), and it is possible to experimentally quantify the period divergence within this model (*Giudicelli et al., 2007*). This also corresponds to the implicit scenario of many theoretical models assuming that the frequency of the clock approaches zero as cells get more anterior, such as the models in *Ares et al., 2012*; *Morelli and Jülicher, 2007*, possibly with a sharp discontinuity suppressing period divergence (*Jörg et al., 2015*; *Soroldoni et al., 2014*). Those models are appealing owing to their simplicity, since all behaviour is encoded in a dynamical frequency gradient (possibly mediated by FGF [*Dubrulle and Pourquié, 2004*]). However, it is unclear what happens from a dynamical systems theory standpoint (a noteworthy exception being the proposal that the gradient emerges through a singularity in phase similar to the Burger's equation [*Murray et al., 2013*]). In particular, the pattern in this scenario literally corresponds to a frozen clock, such that there is an infinite number of local steady states corresponding to the frozen phases of the oscillators.

A second scenario grounded in dynamical systems theory has been proposed (*François and Siggia, 2012*). In this scenario, a genetic network transits from an oscillatory state to an ensemble of (stable) epigenetic states (in Waddington's sense) fixing the pattern. Possible examples include the initial reaction-diffusion based model by *Meinhardt, 1986*, or the cell-autonomous model under morphogen control evolved in *François et al., 2007*; *Figure 1G*. Based on geometric arguments, if bifurcations are local, the most generic model of this transition is expected to present two steps as explained in *François and Siggia, 2012*. As a steep control parameter (possibly controlled by a morphogen such as FGF) decreases, the oscillation dies out through a Hopf bifurcation, leading to a single transient intermediate state. Then, for even lower values of the morphogen, one or several new (stable) states appear (technically through saddle-node bifurcations, see *Figure 1—figure supplement 1*). If the system translates rapidly enough from the oscillatory regime to the multistable regime, a pattern can be fixed (*Figure 1H–K*). Contrary to the previous scenario where the period of the clock goes to infinity, a Hopf bifurcation is associated to a finite period when the clock stops. The pattern of gene expression itself is laid thanks to multiple expression states discretizing the phase of the clock (*Figure 1—figure supplement 1*). Importantly, a finite number of states are observed, for example anterior and posterior fates within one somite (as first pointed out by *Meinhardt, 1982*).

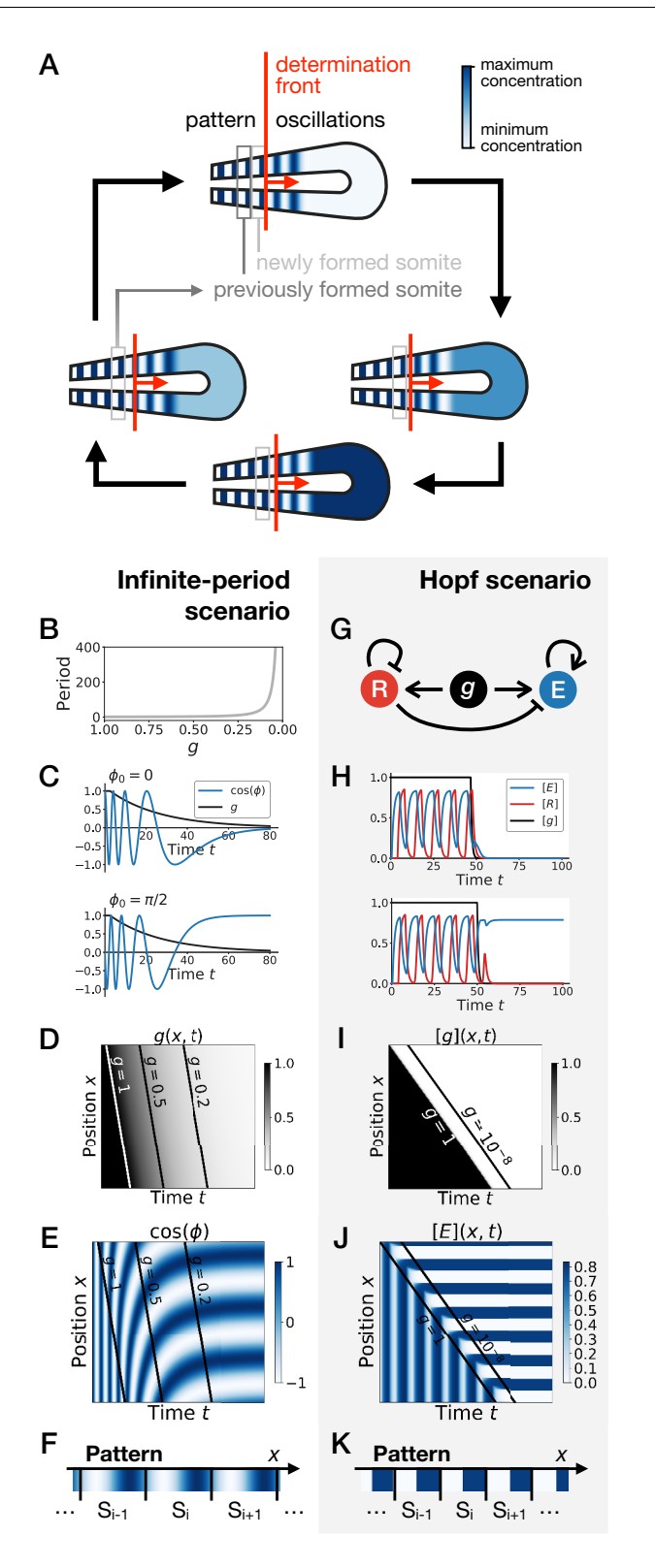

**Figure 1.** Scenarios for segment formation. (**A**) General phenomenology of segment or somite formation. The white to blue gradient represents the oscillating system (e.g. some Notch signaling pathway gene). The determination front (red vertical line) sweeps the embryo in the posterior direction (red arrow) and translates the periodic expression of a genetic clock into a spatial pattern. (**B–F**) Pattern formation with the infinite-period scenario. (**B**) Period divergence is imposed as control parameter $g$ decreases from 1 to 0. (**C**) Two simulated cells with the same dynamics of $g$ end up with different

*Figure 1 continued on next page*

*Figure 1 continued*

final values of the phase. (D–E) Kymographs showing respectively the dynamics of parameter $g$ used in the simulated embryo and the dynamics of the genetic clock. (F) Schematic of the final pattern. (G–K) Pattern formation with the Hopf scenario. (G) Schematic of the gene regulatory network. (H) Depending on the dynamics of $g$, simulated cells can end up with either a high or a low concentration of protein $E$. (I–J) Kymograph showing respectively the dynamics of parameter $g$ used in the simulated embryo and the dynamics of protein $E$. (K) Schematic of the final pattern. The boundary between two segments ('$S_i$') is set arbitrarily at the transition from high to low concentrations of protein $E$.

The online version of this article includes the following figure supplement(s) for figure 1:

**Figure supplement 1.** Bifurcation analysis of the Hopf scenario of *Figure 1*.
**Figure supplement 2.** Two-enhancer model for *Tribolium* segmentation.

In this paper, we revisit those ideas with explicit modeling to characterize the behavior of systems transitioning from a dynamical regime (such as an oscillation) to a static multistable regime. We introduce two new assumptions: 1. the two different phases of developmental expression (dynamic and static) can be separated into two independent sets of transcriptional modules acting on several genes simultaneously, and 2. the system smoothly switches from one set to the other. This proposal is motivated by the recent suggestion in insects that different sets of enhancers control waves of gap genes at different phases of embryonic growth (*El-Sherif and Levine, 2016*). Such assumptions explain the so-called 'speed-gradient' model suggested to explain the gene expression wave dynamics observed during AP patterning in the beetle *Tribolium* (*Zhu et al., 2017*) (see *Figure 1— figure supplement 2*) and (with some additional assumptions) the more subtle gene expression dynamics observed during AP patterning in *Rudolf et al., 2020*; *El-Sherif and Levine, 2016*. Using both gene-network and geometric formalisms, we characterize the types of bifurcations found in systems transitioning from a dynamic to a static regime. Surprisingly, we find that if the transition is smooth enough, global bifurcations appear. This situation is different from the standard scenario (Hopf and saddle-nodes) that we nevertheless recover if the transition is more non-linear. This is a generic result that is better studied and understood using geometric models. We further show that the transition through a global bifurcation is more robust than the sequence of Hopf and saddle-node bifurcations with respect to several perturbations that we simulate. Finally, we find that this model can explain many features of metazoan segmentation, such as 'speed-gradient' mechanisms or changes of spatial wave profiles due to relaxation-like oscillations, and we discuss biological evidence and implications. This geometric approach thus offers a plausible scenario underlying embryonic patterning with many associated experimental signatures.

## Model

In the following, we consider a class of developmental models based on the combination of (at least) two different transcriptional modules. Biologically, those two modules correspond to two sequential developmental phases. The main assumptions are that those transcriptional modules are globally regulated for multiple genes at the same time (which could be done for instance through chromatin global regulations) and that there is a continuous transition from one to the other. Here, we focus on metazoan segmentation and regionalization, but the formalism might be applicable to other patterning processes where both an initiation phase and a maintenance phase have been described.

We use ordinary differential equations to model our system. Calling $P$ a vector encoding the state of all proteins in any given cell (typically $P$ corresponds to concentrations of proteins), a generic single-cell equation describing all models presented in the following is:

$$\dot{P} = \theta_D(g)\,D(P) + \theta_S(g)\,S(P) + C(P) + \eta(g, P) \tag{1}$$

In *Equation 1*, variable $g$ encodes an external control parameter of the developmental transition. For example, $g$ could be an external morphogen concentration driving patterning, but more complex situations with feedback are possible, where $g$ could also be part of the system (e.g. the phase difference between oscillators [*Beaupeux and François, 2016*; *Sonnen et al., 2018*]). For simplicity, we rescale variables so that $g$ is constrained between 0 and 1. The terms $D(P)$ and $S(P)$ correspond to different sets of modules, their influence on the dynamics being weighted by functions $\theta_D(g)$ and $\theta_S(g)$, respectively. The term $\eta(g, P)$ encodes the noise. Finally, $C(P)$ represents dynamical terms that are independent of the transcriptional modules, such as protein degradation.

We focus here on the simplest two-module case, where $S(P)$ encodes a multistable system (i.e. with multiple fixed points at steady state) and $D(P)$ a dynamic system (i.e. oscillatory). In this situation we will assume $\theta_S(0) = 1$, $\theta_S(1) = 0$, $\theta_D(0) = 0$, and $\theta_D(1) = 1$, meaning that for $g = 1$ the network is in a pure dynamic phase, while for $g = 0$ the network is multistable. Details on the specific forms of $D(P)$, $S(P)$, $\theta_D(g)$ and $\theta_S(g)$ are given in the following and in the Appendix. We study two types of models: gene-network like models where $D(P)$ and $S(P)$ explicitly model biochemical interactions between genes (such as transcriptional repression), and geometric models where $D(P)$ and $S(P)$ directly encode flows in an abstract 2D phase space, similarly to *Corson and Siggia, 2017*.

We model an embryo as a line of cells, corresponding to the antero-posterior axis. The dynamics within each cell (position $x$) is described by *Equation 1*. The only difference between cells is that the dynamics of $g$ is a prescribed function of $x$, for example we assume that there is a function $g(x, t)$ describing the dynamics of a morphogen. We focus on the transition between the two regimes as $g$ continuously changes from 1 to 0 in different cells as a function of time. We will typically consider a sliding morphogen gradient moving along the antero-posterior axis with speed $v$, described by $H(s(x - vt))$ where the function $H$ encodes the shape of the morphogen, and parameter $s$ is a measure of the gradient's spatial steepness.

We also include noise in the system with the help of an additive Gaussian white noise. For gene networks, we follow an approach similar to the τ-leaping algorithm (*Gillespie, 2001*), where the variance of the noise corresponds to the sum of the production and the degradation terms (approximating independent Poisson noises). A multiplicative noise intensity term $\sqrt{1/\Omega}$ is introduced, where $\Omega$ can be interpreted as the typical concentration of the proteins in the system, so that bigger $\Omega$ corresponds to lower noise. In addition, we add diffusion coupling the cells in the stochastic gene network models. For the geometric model, the variance of the noise is held independent of the position $x$. A more detailed description of the noise and diffusion terms is provided in the Appendix.

All source codes and data used for this paper are available at: https://github.com/laurentjutras-dube/Dual-Regime_Geometry_for_Embryonic_Patterning (copy archived at https://github.com/elifesciences-publications/Dual-Regime_Geometry_for_Embryonic_Patterning).

## Relation to existing biological networks

The model described above aims at being generic, but one can relate it to existing developmental networks. A two modules dichotomy was initially proposed in the *Drosophila* context, where two enhancers (early and late) were observed for *Krüppel* and *knirps* (*El-Sherif and Levine, 2016*). An extension of this model has been proposed in *Zhu et al., 2017* for a gap gene cascade under control of the maternal gene *Caudal*, which would thus play the role of $g$ (a detailed model is reproduced in the Supplement, see *Figure 1—figure supplement 2*). The dynamic module corresponds to a genetic cascade comprising *hunchback, Krüppel, mille-pates,* and *giant.* Each gene in this sequence activates the next one, and later genes repress earlier ones. For the static module, each gene self-activates, and *hunchback* and *Krüppel* repress one another. The situation is less clear for vertebrates, given the plethora of oscillating genes and possible redundancy in three different pathways, specifically *Notch, Wnt* and *FGF* (*Dequéant et al., 2006*). The two-module system we consider assumes both the dynamic and static regimes are realized by the same set of genetic components (genes and/or signalling pathways). *Notch* signaling is an ideal candidate to be a component of both a dynamic and a static regime that might mediate vertebrate somitogenesis, since Notch is implicated in both the core segmentation clock of vertebrates (e.g. via *Lfng* [*Dale et al., 2003*], or the *hes/her* family in zebrafish [*Lewis, 2003*]) and oscillation stabilization (*Jiang et al., 2000*; *Oginuma et al., 2010*). Importantly, several genes of the Notch signalling pathways (e.g. *DeltaC* in zebrafish) are first expressed in an oscillatory manner then stabilize in striped patterns, as expected in our model (*Giudicelli and Lewis, 2004*; *Wright et al., 2011*). There are also multiple genetic interactions between members of the *Notch* pathway, in particular again *Delta* genes (*Schwendinger-Schreck et al., 2014*), with different roles and changes of regulations in the dynamic vs static phase (see e.g. [*Wright et al., 2011*]).The oscillation itself could be mediated through one or several negative feedback loops in this pathway (*Lewis, 2003*), and stabilization could be realized through one of the multiple *Notch* signaling interactions (possibly via cell coupling, similarly to what is observed in other systems [*Corson and Siggia, 2017*]).

## Results

### A model for the transition between two genetic modules: Hopf vs. SNIC

In *Zhu et al., 2017*, it was suggested that the transition from a 'wave-like' behaviour to a static pattern during *Tribolium* segmentation was mediated by a smooth transition from one set of modules (corresponding to the oscillatory phase) toward another (corresponding to the fixed pattern). This explained the 'speed-gradient' mechanism where the typical time-scale of the dynamical system depends strongly on an external gradient (in this case, the concentration of the transcription factor *Caudal*). In the Appendix, we further study the associated bifurcation, and observe that new fixed points corresponding to the stabilization of gap gene expressions appear on the dynamical trajectory of those gap genes (*Figure 1—figure supplement 2F*). In simple words, the gap gene expression pattern slowly 'freezes' without any clear discontinuity in its behaviour from the dynamic to the static phase, which is reminiscent of the 'infinite-period' scenario displayed on *Figure 1*.

We first aim to generalize this observed property. A simple way to generate waves of gene expressions (as in the gap-gene system described above) is to consider an oscillatory process, so that each wave of the oscillation corresponds to a wave of gap genes. We are not saying here that the gap-gene system is an oscillator, but rather that its dynamics can be encompassed into a bigger oscillator (*Verd et al., 2018*). The other advantage of considering oscillators is that we can better leverage dynamical systems theory to identify and study the bifurcations. Furthermore, it allows for a better connection with oscillatory segmentation processes in vertebrates and arthropods.

We thus start with an idealized gene regulatory network with three genes under the control of two regulatory modules (*Figure 2*). In the dynamic phase $D(P)$, we assume that the three genes are oscillating with a repressilator dynamics (*Elowitz and Leibler, 2000*), so that the system keeps a reference dynamical attractor and an associated period. In the static phase $S(P)$, we assume that the module encodes a tristable system via mutual repression (*Figure 2A*).

We study the dynamics in a simulated embryo under the control of a regressing front of $g$ (*Figure 2B*). Transition from the dynamic module to the static module is expected to form a pattern by translating the phase of the oscillator into different fates, implementing a clock and wavefront process similar in spirit to the one in *François et al., 2007*. We compare two versions of this model, presenting the two different behaviors that we found. In Model 1 (*Figure 2C–H*), the weights of the two modules are non-linear in $g$: $\theta_D(g) = g^2$ and $\theta_S(g) = (1 - g)^2$ (*Figure 2C*). In Model 2 (*Figure 2I–N*), the weights of the two modules are linear in $g$: $\theta_D(g) = g$ and $\theta_S(g) = 1 - g$ (*Figure 2I*). We note that the initial and final attractors of both models are identical. Importantly, only the *transition* from one set of modules (and thus one type of dynamics) to the other is different. This two-module system thus offers a convenient way to compare the performance of different modes of developmental transition while keeping the same 'boundary conditions' (i.e. the same initial and final attractors).

*Figure 2E* and *Figure 2K* show the kymographs for both models without noise, with behaviors of individual cells in *Figure 2D* and *Figure 2J*. While the final patterns of both models are the same (*Figure 2F* and *Figure 2L*), giving rise to a repeated sequence of three different fates, it is visually clear that the pattern formed with Model 2 is more precise and sharper along the *entire dynamical trajectory* than the one formed with Model 1, which goes through a 'blurry' transitory phase (compare mid-range values of $g$ on *Figure 2E* and *Figure 2K*).

To better understand this result, we plot the bifurcation diagram of both models as a function of $g$ in *Figure 2G* and *Figure 2M*. As $g$ decreases, Model 1 is the standard case of a local Hopf bifurcation (*Strogatz, 2015*), which happens at $g = 0.72$. Three simultaneous saddle-node bifurcations appear for lower values of $g$, corresponding to the appearance of the fixed points defining the three regions of the pattern. The behaviour of Model 2 is very different: the fixed points form on the dynamical trajectory, via three simultaneous Saddle Node on Invariant Cycle (or SNIC) bifurcations (*Strogatz, 2015*). Both models display waves corresponding to the slowing down of the oscillators, leading to a static regime. In Model 1, the time-scale disappears with a finite value because of the Hopf bifurcation (*Figure 2H*). For Model 2, it diverges because of the SNIC (*Figure 2N*), suggesting an explicit mechanism for the infinite-period scenario of *Figure 1*.

To further quantify the differences of performance between the two models, we introduce noise (encoded with variable Ω, see the Model section and the Appendix) and diffusion (*Figure 3A–D*).

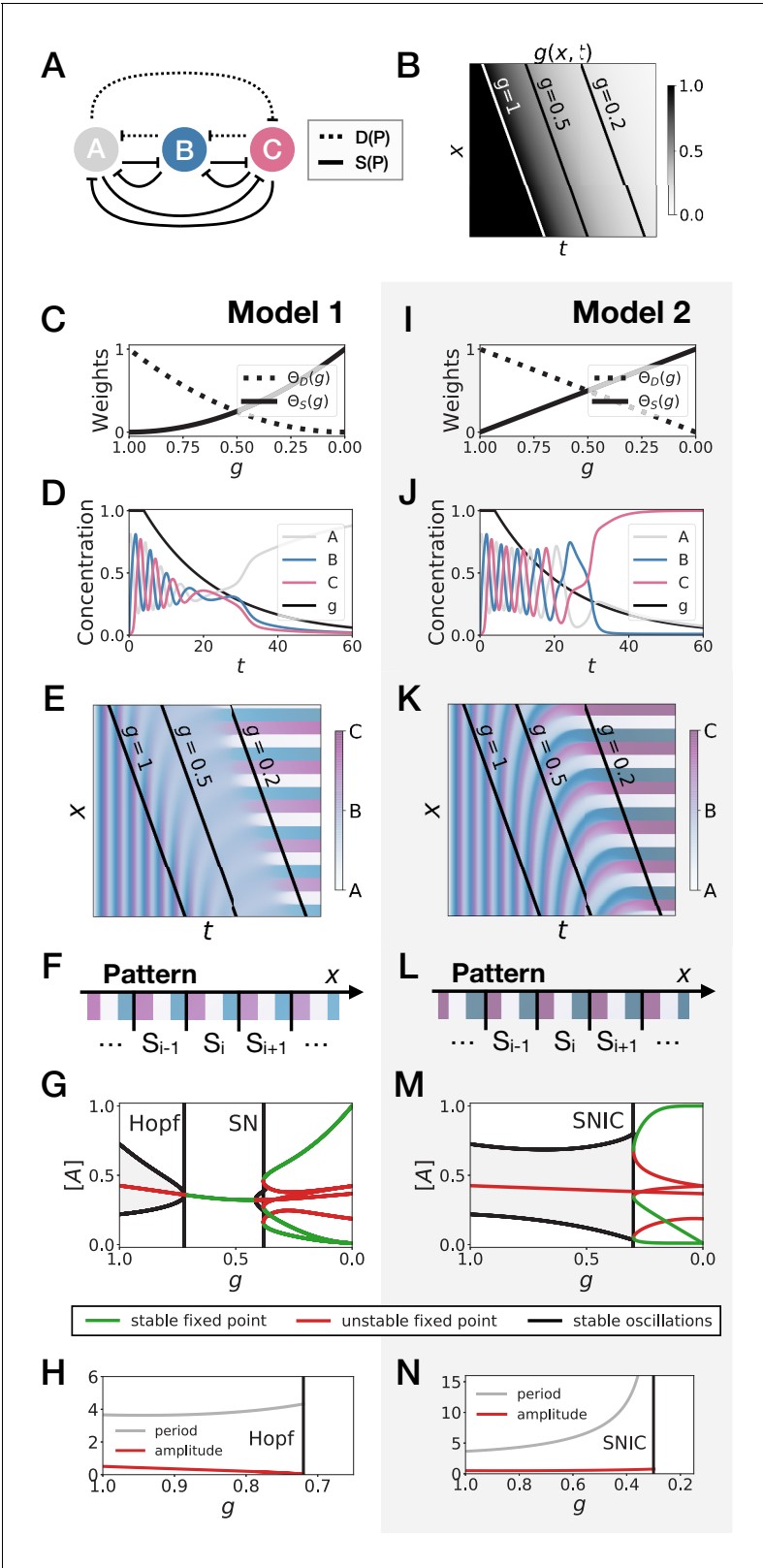

**Figure 2.** 3-gene models for pattern formation. (**A**) Schematic of the gene regulatory networks encoded by the dynamic term (dotted line) and the static term (solid line). (**B**) Kymograph showing the dynamics of parameter $g$ used in the simulated embryos for both Models 1 and 2. (**C–H**) Simulation results for Model 1. (**C**) Weights of the dynamic (dotted line) and static (solid line) modules as a function of parameter $g$. (**D**) Gene concentration and value of parameter $g$ inside a representative simulated cell as a function of time. (**E**) Kymograph showing the dynamics of gene expression in the

*Figure 2 continued on next page*

*Figure 2 continued*

simulated embryo. Transparent colors are used to represent the concentration of the three genes, so that mixes of the three genes can be easily perceived. Genes *A*, *B*, and *C* are shown in transparent white, blue and purple, respectively. Simulated cells with intermediate concentrations of all genes appear grey. (F) Schematic of the final pattern. (G) Bifurcation diagram showing the types of dynamics available to the simulated embryo as a function of parameter *g*. The maximum and minimum concentrations of gene *A* on the stable limit cycles are shown in black. Stable and unstable fixed points are shown in green and red, respectively. 'SN' stands for saddle-node bifurcation. (H) Period (grey line) and amplitude (red line) of the oscillations along the stable limit cycle. (I–N) Simulation results for Model 2.

The online version of this article includes the following figure supplement(s) for figure 2:

**Figure supplement 1.** 3-gene models for pattern formation with Hill functions for the weights.

**Figure supplement 2.** Peak-to-peak frequency in the 3-gene models.

We also define a mutual information metric measuring how precisely the phase of the oscillator is read to form the final pattern (*Figure 3E*, see the Appendix for details), consistent with the experimental observation in vertebrate segmentation that oscillatory phases and pattern are functionally connected (*Oginuma et al., 2010*). Intuitively, this metric quantifies in a continuous way the number of fates encoded by the system at steady state. Ideal mutual information for the three mutually exclusive genes of Models 1 and 2 gives $log_2(3) \sim 1.6$ bits of mutual information, meaning that the pattern deterministically encodes the phase of the cycle into three static fates with equal weights. While addition of noise decreases this mutual information as expected (*Figure 3E*), Model 2 (black curves) always outperforms Model 1 (red curves). For a reasonable level of noise corresponding to a few thousands of proteins in the system, Model 2 can encode $2^{1.3} \sim 2.5$ fates, close to the optimum 3. Furthermore, for a given diffusion constant, Model 1 requires a ten times smaller noise level than Model 2 to encode the same amount of mutual information, which thus indicates much better noise resistance for Model 2.

Those observations suggest that appearance of stable fixed points through SNIC bifurcations rather than through Hopf bifurcations generates a more robust pattern. The superiority of Model 2 can be rationalized in the following way: when there is a Hopf bifurcation, only one fixed point exists for a range of *g* values, so that all trajectories are attracted towards it. This corresponds to the 'blurred' zone in the kymographs of *Figure 2* and *Figure 3*. In presence of noise, the effect is to partially erase the memory of the phase of the oscillation when only one fixed point is present for the dynamics. Conversely, a SNIC bifurcation directly translates the phase of the oscillation into fixed points, without any erasure of phase memory, ensuring higher information transfer from the dynamic to the static phase, and therefore more precise patterning.

The Hopf bifurcation of Model 1 occurs when the weights of the dynamic and static modules become small compared to the degradation term, which generates an 'intermediate regime' with one single fixed point after the oscillations of the dynamic module and before the multistability of the static module. The specific form of the weights is not what determines the bifurcation, but rather the presence or absence of an intermediate regime. We confirm this observation with similar 3-gene models that used Hill functions for the weights $\theta_D$ and $\theta_S$ (*Figure 2—figure supplement 1* and *Figure 3—figure supplement 1*). Interestingly, we get both Hopf and SNIC bifurcations with the same shape for the two weights; the Hopf is obtained by shifting the weight of the dynamic term toward larger values of the control parameter. This effectively generates the required intermediate regime where both weights are small compared to the degradation term.

## Gene-free models present a similar geometry of transition

Hopf and saddle-node bifurcations are 'local' bifurcations, in the sense that changes of the flow in phase space are confined to an arbitrarily small region of phase space as the bifurcation is approached. They do not in principle require complex changes of the flow or fine-tuning of the parameters to happen. As such, they are the most standard cases in many natural phenomena and in most theoretical studies. Conversely, SNIC bifurcations are 'global' bifurcations (*Strogatz, 2015*; *Ermentrout, 2008*): they are associated to changes of the flow in large regions of phase space (e.g. when a limit cycle disappears with a non-zero amplitude) and usually require some special symmetries or parameter adjustments to occur (e.g. to ensure that a saddle-node collides with a cycle).

It is therefore a surprise that SNIC bifurcations spontaneously appear in the models considered here. To better understand how this is possible and if this is a generic phenomenon, we follow ideas

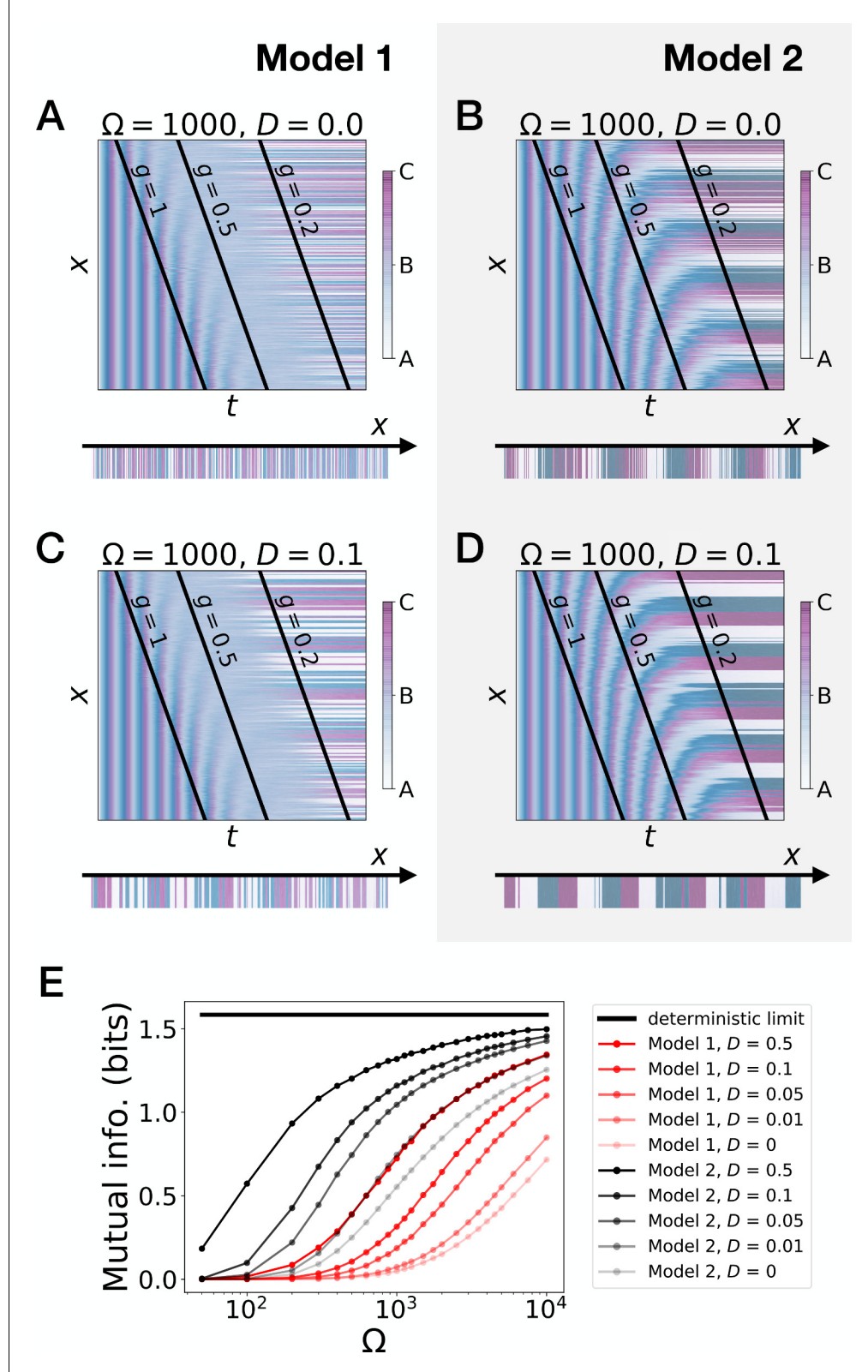

**Figure 3.** Stochastic simulations of the 3-gene models. (A–D) Kymographs showing the stochastic dynamics of gene expression in simulated embryos. The specific values of the typical concentration $\Omega$ and of the diffusion constant $D$ used to generate each kymograph are indicated on the panels. The concentration of the three genes at the last simulated time point is shown schematically in the lower part of each panel. (E) Mutual information as a function of typical concentration $\Omega$ for Model 1 (red lines) and Model 2 (black lines). Paler colors correspond to lower values of the diffusion constant $D$.
*Figure 3 continued on next page*

*Figure 3 continued*

The thick horizontal black line indicates the ideal mutual information for three mutually exclusive genes. Note that higher values of $\Omega$ correspond to lower noise levels.

The online version of this article includes the following figure supplement(s) for figure 3:

**Figure supplement 1.** Stochastic simulations of the 3-gene models with Hill functions for the weights.

first proposed by *Corson and Siggia, 2012* and consider geometric (or gene-free) systems. We aim to see if: 1. SNIC bifurcations are generically observed, and 2. a model undergoing a SNIC bifurcation is in general more robust to perturbations than a model undergoing a Hopf bifurcation, with initial and final attractors being held fixed. We thus build 2D geometric versions of the system (variables $y$ and $z$). The dynamic module $D(P)$ is defined by a non-harmonic oscillator on the unit circle, while the static module $S(P)$ is defined by two stable fixed points, at $y = \pm 1$, $z = 0$ (see *Figure 4A*, and the Appendix for the equations). Like previously, we build a linear interpolation between the two systems as a function of $g$ and explore the consequence on the bifurcations (*Figure 4B–H*). Since the flow in the system is 2D, we can also easily visualize it (*Figure 4I* and *Figure 4—video 2*).

In brief, this geometric approach confirms all the observations made on the gene network model of the previous section, and further clarifies the origin of the SNIC bifurcation. Because of the smooth transition between modules, the entire flow in 2D needs to interpolate from a cycle to a bistable situation. When both modules have close to equal weights (around $g = 0.5$), the flow and associated cycle concentrate around two future fixed points. This appears in retrospect as the most natural way to interpolate between the two situations since both types of attractor (stable limit cycle, and multiple stable fixed points) are effectively present at the same time around $g = 0.5$. For this reason, the oscillations are also more similar to relaxation oscillations, rapidly jumping between two values corresponding to the future fixed points. When $g$ is further lowered, the weight of the static module dominates and 'tears apart' the cycle, forming two fixed points.

This situation is so generic that in fact, to obtain a Hopf bifurcation, we need to mathematically reinforce the fixed point at $y = 0$ for intermediate values of $g$. More precisely, three things are required to get a Hopf bifurcation. First, we need to add an extra term, the 'intermediate module', characterized by a single fixed point at $y = 0$ and $z = 0$. Without this module, we consistently get a SNIC bifurcation, for all forms of coupling tested, including linear and non-linear couplings (*Figure 4—figure supplement 1A–D*). Second, we need to set the weight of the intermediate module to 0 for $g = 1$ and $g = 0$, so that we get the oscillations of the dynamic module at the beginning of the simulation and the bistability of the static module at the end of the simulation. We achieve this by setting the weight of the intermediate term to $g(1 - g)$, which is of second order in $g$. Third, we need to make the weights of the dynamic and static modules smaller than the weight of the intermediate module for intermediate values of the control parameter, that is for $g$ around 0.5. This is achieved by using cubic weights for the dynamic and static modules (*Figure 4—figure supplement 1G* and *Figure 4—figure supplement 2*). Weights of the fourth order in $g$ also lead to a Hopf bifurcation (*Figure 4—figure supplement 1H*), while linear weights lead to a SNIC bifurcation (*Figure 4—figure supplement 1E*). Interestingly, quadratic weights lead to simultaneous supercritical Hopf and pitchfork bifurcations (*Figure 4—figure supplement 1F*). The non-linearity of the coupling helps reinforce the relative weight of the intermediate module for values of $g$ around 0.5, but the exact shape of the non-linearity is not crucial. Furthermore, our mutual information metric confirms that the pattern is more robustly encoded when the system goes through a SNIC bifurcation rather than through a sequence of Hopf and pitchfork bifurcations (*Figure 4—figure supplement 5F*).

Detailed simulations of the gene-free model with the intermediate module and with cubic weights for the dynamic and static modules are shown on *Figure 4—figure supplement 2*. As expected for a Hopf bifurcation, the flow first concentrates on the central fixed point at $y = 0$, before re-emerging in a bistable pattern for lower $g$ (*Figure 4—figure supplement 2I* and *Figure 4—video 1*). There are technically two types of Hopf bifurcations, depending on the stability of the limit cycle. A Hopf bifurcation is supercritical (resp. subcritical) if a stable (resp. unstable) limit cycle becomes a stable (resp. unstable) fixed point. All Hopf bifurcations discussed previously are supercritical. However, by changing slightly the dynamic module of the gene-free model (and including the intermediate

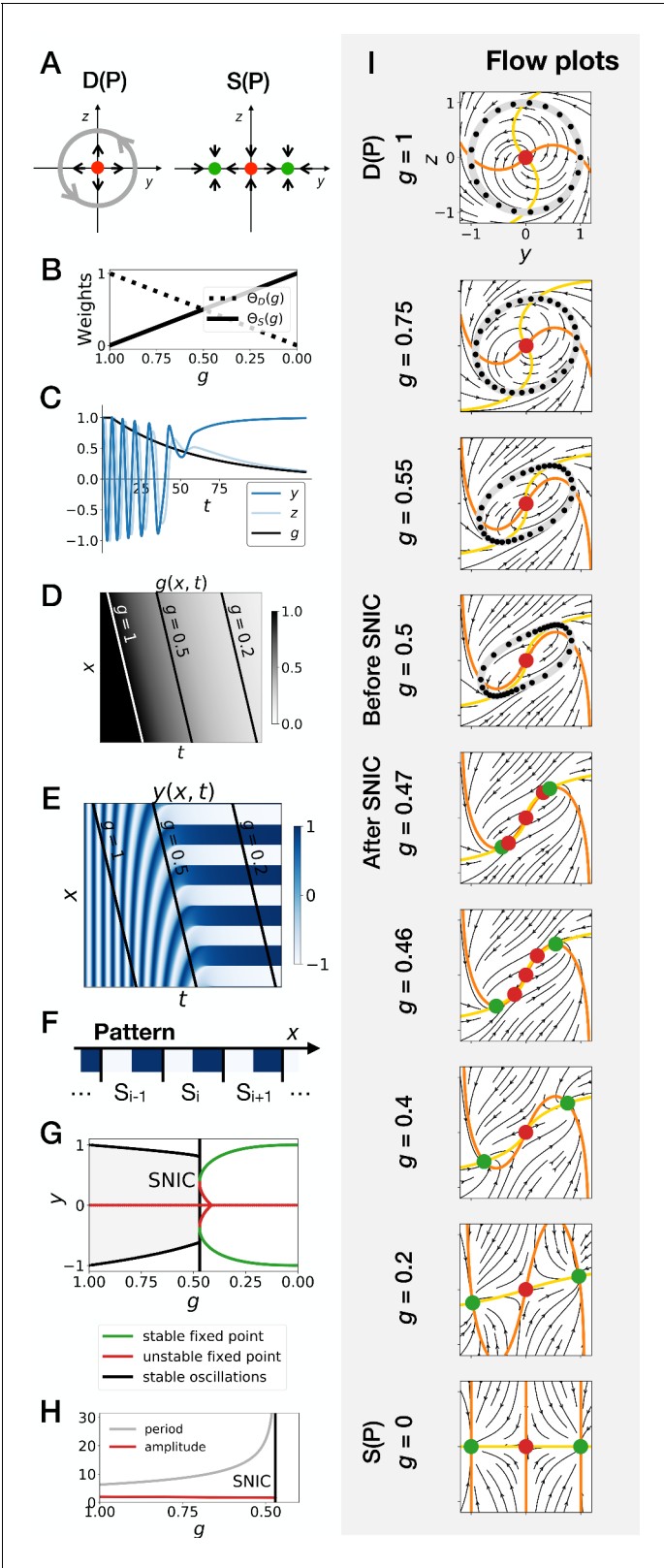

**Figure 4.** Gene-free geometric model for pattern formation (Model 2). (**A**) Schematic of the flow encoded by the dynamic and static terms. The grey circle represents oscillations on the unit circle. Green and red dots represent unstable and stable fixed points, respectively. (**B**) Weights of the dynamic (dotted line) and static (solid line) modules as a function of parameter $g$. (**C**) Values of geometric coordinates $y$ and $z$ and of parameter $g$ in a simulated cell as a function of time. (**D–E**) Kymographs showing respectively the dynamics of parameter $g$ used in the simulated embryo and the dynamics of

*Figure 4 continued on next page*

*Figure 4 continued*

coordinate $y$. (F) Schematic of the final pattern. (G) Bifurcation diagram showing the types of dynamics available to the simulated embryo as a function of parameter $g$. The maximum and minimum values of coordinate $y$ on the stable limit cycles are shown in black. Stable and unstable fixed points are shown in green and red, respectively. (H) Period and amplitude of the oscillations. (I) Flow in phase space for different values of parameter $g$. The same color scheme than panel $A$ is used to represent the cycles and the fixed points. Positions along the limit cycle at time points separated by a fixed time interval are indicated with black dots, so that variations of the speed of the oscillations along the limit cycle can be visualized. The yellow and orange lines represent the $y$ and $z$ nullclines, respectively.

The online version of this article includes the following video and figure supplement(s) for figure 4:

**Figure supplement 1.** Gene-free models with different weights for the dynamic and static modules .
**Figure supplement 2.** Supercritical Hopf scenario in the gene-free model (Model 1).
**Figure supplement 3.** Subcritical Hopf scenario in the gene-free model (Model 3).
**Figure supplement 4.** Gene-free model with two subcritical Hopf bifurcations (Model 4).
**Figure supplement 5.** Stochastic simulations of the gene-free models.
**Figure 4—video 1.** Flow of the gene-free model with a SNIC bifurcation (Model 2).
https://elifesciences.org/articles/55778#fig4video1
**Figure 4—video 2.** Flow of the gene-free model with a supercritical Hopf bifurcation (Model 1).
https://elifesciences.org/articles/55778#fig4video2
**Figure 4—video 3.** Flow of the gene-free model with a subcritical Hopf bifurcation (Model 3).
https://elifesciences.org/articles/55778#fig4video3
**Figure 4—video 4.** Flow of the gene-free model with two subcritical Hopf bifurcations (Model 4).
https://elifesciences.org/articles/55778#fig4video4

module as well as cubic weights for the dynamic and static modules) we can also obtain a subcritical Hopf bifurcation (*Figure 4—figure supplement 3*). During this bifurcation, an unstable limit cycle forms around the origin. This unstable limit cycle coexists with the stable limit cycle of the dynamic module for some range of $g$ values before they annihilate each other during a so-called 'saddle-node of limit cycles' bifurcation. Finally, the two fixed points of the static module form during two simultaneous saddle-node (of fixed points) bifurcations (*Figure 4—figure supplement 3I* and *Figure 4—video 3*). Again, our mutual information metric confirms that the pattern is more precise when the system goes through a SNIC bifurcation (Model 2), rather than through supercritical or subcritical Hopf bifurcations (Models 1 and 3, resp.) (*Figure 4—figure supplement 5E*). Taken together, these results suggest that keeping the static and dynamic attractors fixed, patterning is both more generic and more robustly encoded through a SNIC bifurcation than through a Hopf bifurcation, at least in simple low-dimension models.

## Robustness and asymmetry in the fixed points

A concern with the results of the previous section might be that those mathematical models are in fact fine-tuned and too symmetrical, so that in particular when the transition occurs, both new fixed points appear for the same value of the control parameter. Furthermore, real biological networks have no reason to be perfectly symmetrical (although evolution itself might select for more symmetrical dynamics if needed). We thus relax our hypotheses to study a system where parameters and trajectories are not symmetrical (*Figures 5* and *6*).

Going back first to the gene network model, we induce an asymmetry between the fixed points by changing thresholds of repression in the static phase (*Figure 5A*). The bifurcation diagrams of *Figure 5B–C* indicate that the asymmetry of the fixed points indeed breaks the simultaneity of appearance of all fixed points in both scenarios. We nevertheless notice that for those changes of parameters, all bifurcations still happen in a very narrow range of $g$ for the SNIC model.

Asymmetry of the fixed points might therefore destroy the advantage of SNIC vs Hopf by creating a transient zone where one of the fixed points always dominates. We thus perform a comparison between Models 1 and 2 with the same asymmetric static enhancers (*Figure 5*, see also *Figure 5— figure supplements 1* and *2*, and the Appendix for details). To compare the two cases, we consider different time-scales of the morphogen gradient. The reasoning is that the slower the decay of $g$, the more time the system spends in a region of parameter space without all three final fixed points, allowing the system to relax and 'lose' phase information. Conversely, a faster decay of $g$ means that

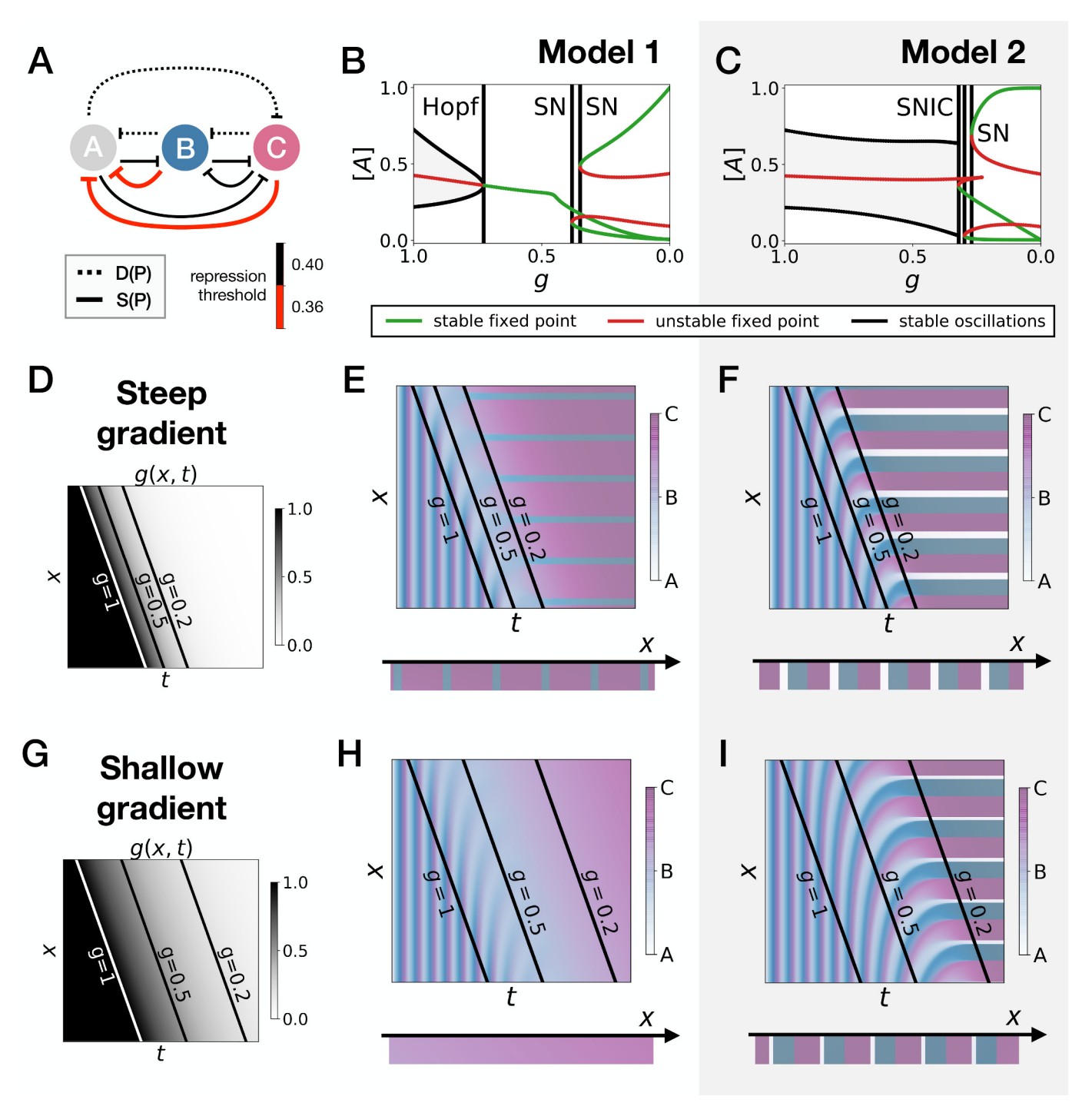

**Figure 5.** Perturbation of the morphogen gradient steepness in asymmetric 3-gene models. (A) Schematic of the gene regulatory networks encoded by the dynamic term (dotted line) and the static term (solid line). The thick red lines indicate stronger repression than the black lines (see the parameters in the Appendix). (B–C) Bifurcation diagram showing the types of dynamics available in Models 1 and 2. The maximum and minimum concentrations of gene *A* on the stable limit cycles are shown in black. Stable and unstable fixed points are shown in green and red, respectively. The main bifurcations are identified with vertical lines. 'SN' stands for saddle-node bifurcation. (D–F) Simulation results for a steep gradient of parameter *g*. (D) Kymograph showing the dynamics of parameter *g* used in the simulated embryos for both Models 1 and 2. (E–F) Kymographs showing the dynamics of gene expression in the simulated embryos of Models 1 and 2. The concentration of the three genes at the last simulated time point is shown schematically in the lower part of the panels. (G–I) Simulation results for a shallow gradient of parameter *g*.

The online version of this article includes the following figure supplement(s) for figure 5:

*Figure 5 continued on next page*

*Figure 5 continued*

**Figure supplement 1.** Perturbation of the morphogen gradient steepness in strongly asymmetric 3-gene models.
**Figure supplement 2.** Perturbation of the morphogen gradient steepness in randomly asymmetric 3-gene models.

less time is spent in a region with few fixed points, and therefore the patterns are expected to be more robust.

We first decrease the thresholds of repression of gene A by both genes B and C (*Figure 5A*). Results of these simulations are shown in *Figure 5*: Model 2 with a SNIC bifurcation still outperforms Model 1 with Hopf and saddle-node bifurcations. In particular, it is again visually clear on kymographs that Model 2 produces a robust and well-defined pattern at any time point of the simulations, while Model 1 gives rise to a much 'fuzzier' pattern before the transition. Model 1 produces a robust static pattern only for a steep gradient (allowing to quickly move through the 'fuzzy' phase) and a weak asymmetry in the static module (*Figure 5E*). It is brittle to any change of the dynamics of $g$ (*Figure 5H*) or to stronger asymmetry in the static module (*Figure 5—figure supplement 1E,H*). Conversely, Model 2 is robust to different shapes of the morphogen (*Figure 5F,I*). Only for a strong asymmetry does the system lose one fixed point (*Figure 5—figure supplement 1I*), but even in this case transitions through a SNIC bifurcation appear superior to transitions through a Hopf bifurcation.

The fragility of the Hopf bifurcation to asymmetries in the parameters can be understood as follows. In the asymmetric versions of Model 1, one of the fixed points of the static term forms at the Hopf bifurcation, way before the two other fixed points form. It is therefore the only attractor available for a large range of $g$ values. However, in Model 2 the same asymmetry only favors one of the fixed points for a small range of $g$ values, generating a robust pattern. Again, we can use the mutual information metric defined above to quantify the robustness of the pattern and confirm the superiority of Model 2 (*Figure 5—figure supplement 2J*). We also confirmed these results for the case of random modifications of the repression thresholds of all interactions in the static term (*Figure 5—figure supplement 2*).

The asymmetry introduced in *Figure 5* changes the shapes of the basins of attraction and the positions of the fixed points. The geometric model allows to change those features independently. The most generic way to introduce an asymmetry in the system is to fix the positions of the fixed points of the static regime and change only the positions of the basins of attraction (the reason is that the future fates depend on the position of the separatrix between different regimes [*Corson and Siggia, 2012*]). To replicate this situation in the 2D gene-free models, we move the unstable fixed point of the static term along the $y$ axis. Results of this procedure are shown on *Figure 6* and confirm our results on the gene-network based models: Model 2 bifurcates via a SNIC and is always more robust than Model 1. When we change the positions of the fixed points in the static regime to move them away from the limit cycle (still in an asymmetric way), interestingly both Models 1 and 2 now bifurcate via SNICs (*Figure 6—figure supplement 1*). Furthermore, we see that for Model 1, the amplitude of the limit cycle decreases before the bifurcation, while for Model 2, the amplitude increases (*Figure 6—figure supplement 1E*).

We conclude from all those numerical perturbations that even with asymmetric basins of attraction and asymmetric parameters, transitions based on SNIC bifurcations are both more generic and more robust than the ones based on Hopf bifurcations, at least in simple low-dimension models.

## Spatial wave profiles: Hopf vs SNIC

Our theoretical study suggest that SNIC based transitions are both more robust and more generic than Hopf/saddle-nodes ones. We thus now examine ways to distinguish between Hopf and SNIC based transitions experimentally. The natural method would be to modify the value of the control parameter of the bifurcations (in our case parameter $g$) and check how the attractors of the system change. A recent experimental example can be found in the auditory hair-bundle context where it has been suggested that Hopf bifurcations can be distinguished from SNIC bifurcations by tuning the external driving force (*Salvi et al., 2016*). However, the situation in development is different from any sensory system since the actual control parameters are not known and are likely combinations of various signaling systems (such as *FGF* or *Wnt*). There could also be many compensatory

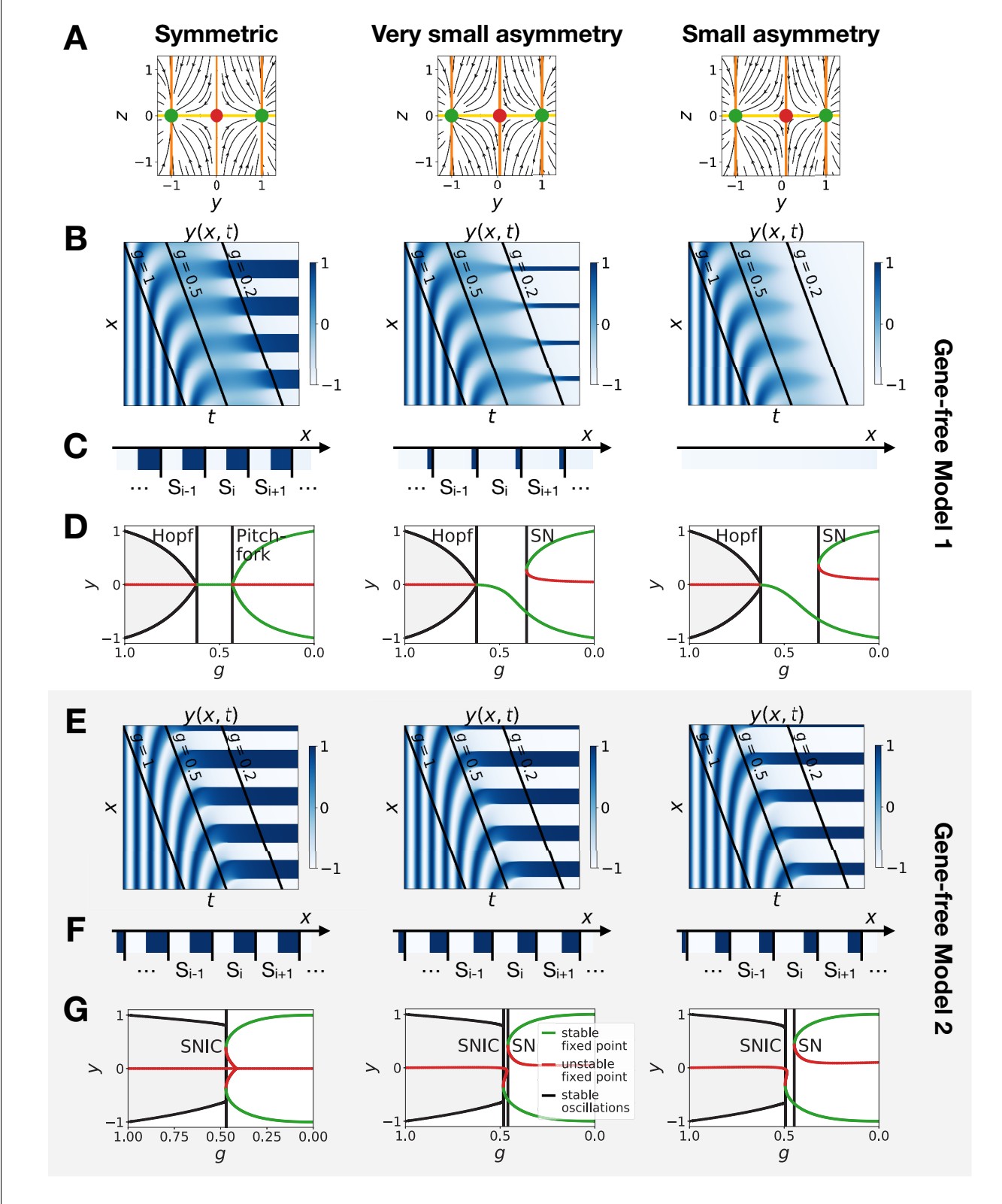

**Figure 6.** Perturbation of the morphogen gradient steepness in the gene-free geometric models. (**A**) Flow plots showing the changes of geometry of the static module. (**B–C**) Corresponding kymographs and final patterns for Model 1. (**D**) Associated bifurcations diagrams. 'SN' stands for saddle-node bifurcation. (**E–F**) Corresponding kymographs and final patterns for Model 2. (**G**) Associated bifurcation diagrams.

*Figure 6 continued on next page*

*Figure 6 continued*

The online version of this article includes the following figure supplement(s) for figure 6:

**Figure supplement 1.** Model one becomes a SNIC when fixed points are outside the limit cycle.

developmental mechanisms, making it difficult to fully take control of the system, as well known by developmental geneticists.

In the absence of a direct control, we have to rely on indirect measurements. The most obvious choice is to use the antero-posterior position along the embryo (or along the PSM in the case of somite formation) as a proxy for the control parameter. In situs are informative of what locally happens in insects (*El-Sherif et al., 2012*; *Zhu et al., 2017*). In vertebrates, it is possible to monitor in real-time segmentation oscillators in embryos (*Aulehla et al., 2008*; *Webb et al., 2016*; *Delaune et al., 2012*) and in tissue cultures (*Lauschke et al., 2013*; *Matsuda et al., 2020*; *Diaz-Cuadros et al., 2020*), so that many properties of the cycles can be inferred.

A first relevant metric (used in *Salvi et al., 2016*) is the shape of the limit cycle as a function of the control parameter. There is a clear contrast between Hopf and SNIC bifurcations on our simulated kymographs: for Hopf bifurcations, as mentioned above the transition zone is 'blurred' because the damped oscillations relax toward the unique fixed point, while for SNIC bifurcations the oscillations localize close to the fixed points, and as a consequence the pattern is gradually reinforced (*Figures 2*, *3*, *4*, *5*; damped oscillations are especially visible in the geometric model). To better visualize the 'blurry' transition zone characteristic of Hopf bifurcations in the gene-free model (resp. in the 3-gene model), we compute the distribution of the values of the geometric variables (resp. of the gene concentrations) for different values of parameter $g$ (*Figure 7—figure supplement 2*). In models with Hopf bifurcations, the distribution becomes very narrow for intermediate values of $g$, which is a direct consequence of the damped oscillations past the Hopf bifurcations. This suggests that measuring experimentally the distribution of gene concentrations across developmental time at fixed antero-posterior positions can help distinguish SNIC from Hopf bifurcations during actual pattern formation processes. The experimental picture, both from in situ and live imaging, does not support a narrowing of the distribution: it shows clear increases of amplitude that seems more consistent with a gradual reinforcement of the pattern, similar to a SNIC (see e.g. [*Delaune et al., 2012*]).

Another metric that could help identify the type of bifurcation is the time evolution of the period (resp. of the frequency), which presents a discontinuity for both types of Hopf bifurcations, but continuously goes to infinity (resp. zero) for SNIC bifurcations. The local spatial wavelength of the pattern provides a continuous measurement related to the local period (as first used and derived in *Giudicelli et al., 2007*). Calling $S$ the somite size (or the wavelength of anterior/posterior markers within somites after clock stopping), $T(x)$ and $S(x)$ the respective period and local wavelength of the pattern at position $x$ (with $x = 0$ being the tail and $x = 1$ the front), and assuming the cells move towards the anterior with constant speed, we have

$$S(x) = \frac{S}{1 - \frac{T(0)}{T(x)}}$$

In the posterior $T(x) = T(0)$, so that $S(x)$ is infinite: all cells are synchronized locally. As cells move towards the anterior, $T(x)$ decays and $S(x)$ decreases but stays bigger than $S$. If there is a Hopf bifurcation, one expects that the clock will stop with a final period $T(1)$, corresponding to a critical wavelength of the wave $S(1) = S \frac{T(1)}{T(1) - T(0)}$, which is strictly greater than $S$. There is therefore in principle a discontinuity between the wavelength of the propagating wave in the PSM and the wavelength of the static pattern. Conversely, if $T(x)$ goes continuously to infinity, $S(x)$ decays continuously to converge towards $S$ (as measured in *Giudicelli et al., 2007*). This is a simple and intuitive experimental outcome: in this situation the wavelength of the propagating wave in the PSM decreases until it exactly matches the pattern of anterior markers. Therefore, a signature of the finite vs infinite period bifurcation might be visible by comparing the local wavelength of propagating waves in the PSM to the wavelength of anterior markers in the pattern. Practically, monitoring the distance between the peaks of the oscillations does not give a clear difference between Hopf and SNIC, the reason being

that such a discrete measurement can smooth out an abrupt change in the period and that past the Hopf bifurcation, damped oscillations actually give rise to a 'ghost' frequency which can decrease quite significantly (see *Figure 2—figure supplement 2*). Therefore, a measurement of the local wavelength might be less conclusive than a measurement of the amplitude. Nevertheless, we notice that a SNIC scenario gives rise to a combination of an increase of the amplitude with a continuous shrinking of the wavelength, meaning that many refining waves (corresponding to broad variations of periods) will simultaneously propagate.

Finally, SNIC bifurcations are also accompanied by characteristic changes of the shape of spatial wave profiles that might be observable in experiments. We further compare a SNIC-based model to a phase model where an infinite-period behaviour is explicitly assumed, namely the model of a collection of coupled oscillators from *Morelli et al., 2009*. A kymograph of the spatio-temporal profile of the frequency imposed on the oscillators is shown in *Figure 7A*, and the dynamics of the resulting pattern formation process is shown on the kymograph of *Figure 7B*, with the final pattern on *Figure 7C*. The most striking difference is observed on the shape of the spatial wave profile as it moves towards the region where the pattern stabilizes. In the infinite-period scenario of

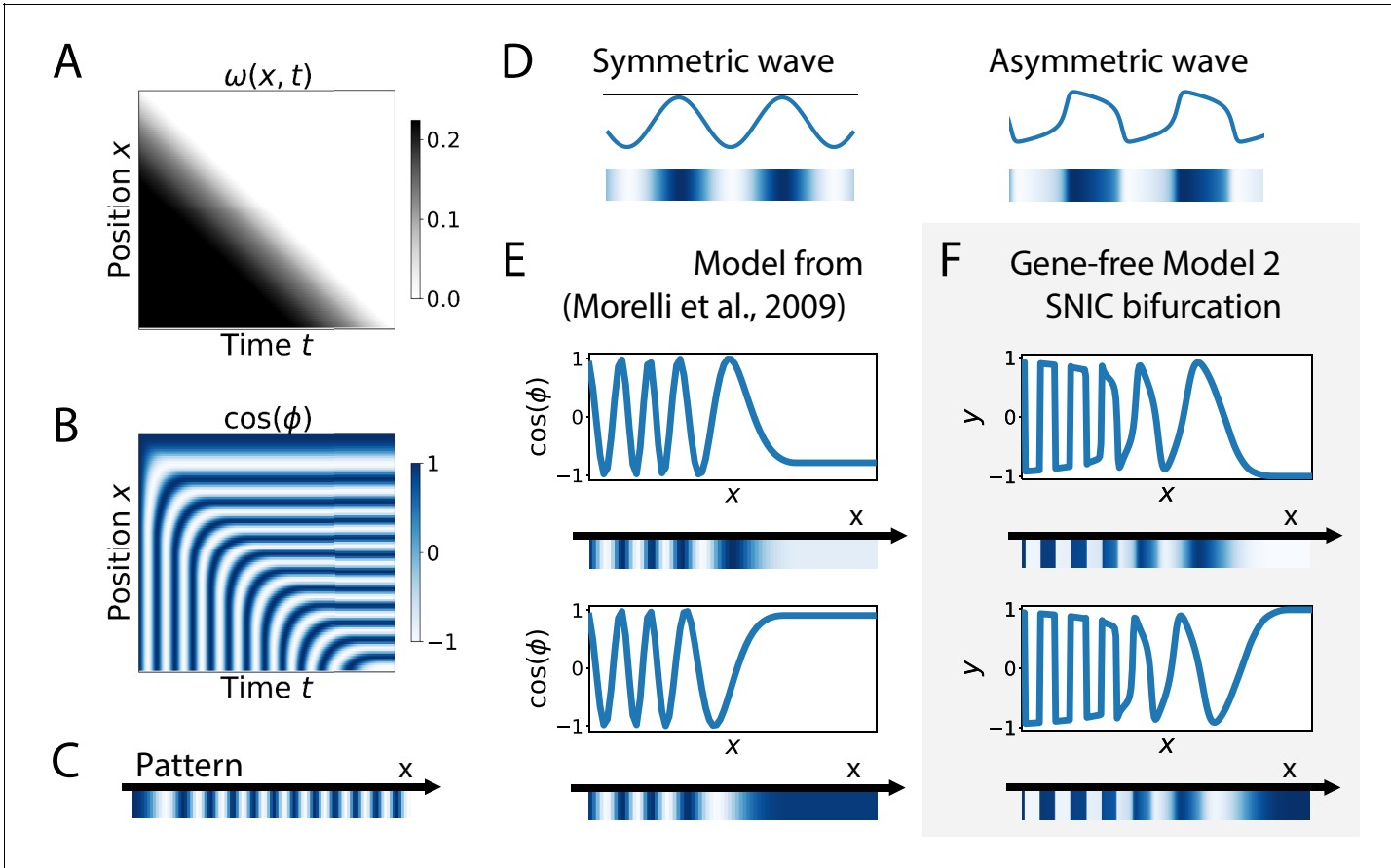

**Figure 7.** Wave pattern in different models for the infinite-period scenario. (A) Frequency profile for the simulation of the model of coupled oscillators from *Morelli et al., 2009*. (B–C) Kymograph showing the dynamics of the phase of the oscillators and the corresponding final pattern. (D) Two schematic examples of possible wave patterns (symmetrical vs asymmetrical). The symmetric wave is obtained with a sine function. The asymmetric wave is retrieved from simulations of a Van der Pol oscillator. (E) Wave pattern for the model of Panels (A–C) for two different time points. (F) Wave pattern for Model 2 of *Figure 4* for two different time points.

The online version of this article includes the following video and figure supplement(s) for figure 7:

**Figure supplement 1.** Wave pattern in different versions of the 2D gene-free model.

**Figure supplement 2.** Distribution of the values of the geometric variables (gene-free models) and of the gene concentrations (3-gene models).

**Figure 7—video 1.** Comparison of pattern formation dynamics in different models.

https://elifesciences.org/articles/55778#fig7video1

*Morelli et al., 2009*, the phase profile is by construction symmetric (albeit stretched in the posterior compared to the anterior, see *Figure 7E*). In the SNIC scenario, we see a clear asymmetry in the peaks of the wave profile (as shown schematically in *Figure 7D*). Indeed, following the peaks of oscillations spatially from anterior to posterior (left to right), we see that the transition from positive to negative values of $y$ occurs in two steps. During the first step, the system stays close to $y = 1$ for some time, with $y$ values decreasing slowly, while in the second step the system goes rapidly towards negative $y$ values and reaches $y = -1$ fast (*Figure 7F*). This phenomenon, which gives rise to a 'sawtooth' pattern of propagating waves, is observed in all our versions of Model 2 (and is notably absent from all our versions of Model 1, see *Figure 7—figure supplement 1*). Those spatial asymmetries are likely due to the asymmetries of the limit cycles of relaxation oscillators, where a system jumps between two (or more) steady states in an asymmetric way (which can also be observed in systems close to criticality, see *Tufcea and François, 2015*). Our model thus offers a simple explanation of wave asymmetry, solving the long-standing problem of the asymmetry of AP vs PA transitions, which is possibly crucial for segment polarity as first suggested by *Meinhardt, 1982*.

## Discussion

In this work, we have explored the dynamical properties of generic two-module systems, where one module corresponds to a dynamic phase of genetic expression and the other corresponds to a static phase controlling embryonic patterning. The unexpected result is that those models typically present global bifurcations where new fixed points appear on the trajectories in phase space. Such SNIC bifurcations come from the smooth interpolation between a flow defining an oscillator in phase space and a landscape characterized by several fixed points. The oscillating attractor then gets continuously deformed until it breaks into several fixed points, leading to the SNIC. This interpolation is a direct consequence of the assumed two-module control as shown on multiple examples above. Importantly, the overall developmental sequence in this context is emergent, since the dynamics close to the bifurcation cannot be understood independently from the static or dynamic modules only. SNIC bifurcations also provide robustness to various perturbations (since fixed points appearing on cycles better preserve information on the oscillatory phase). Below we detail how this model also recapitulates many observed features of metazoan segmentation.

### Experimental evidence

In absence of a direct handle on the control parameter, comparison of SNIC and Hopf bifurcations in simulated systems reveal three possible signatures of SNIC bifurcations in the propagating antero-posterior waves of genetic expression: 1. oscillations gradually reinforcing the pattern for SNIC (instead of damped oscillations for Hopf) 2. relaxation-like oscillations leading to some asymmetry in the wave pattern, and 3. a continuous shrinking of the spatial wavelength during the dynamic phase to match the static pattern for SNIC, with many well-defined propagating waves.

For prediction 1, decades of in situs of oscillating genes and of experimental monitoring of *Notch* signalling pathways argue against damped oscillations in somitogenesis. Weak waves gradually refine into well-defined stripes in the anterior PSM, in most species (see for example the comparison made in *Gomez et al., 2008*), suggesting an increase of amplitude in oscillations. Consistent with this, visualising segmentation clock oscillations in live vertebrate embryos suggests an increase of oscillation amplitude as cells get more anterior in the PSM (*Delaune et al., 2012*; *Lauschke et al., 2013*). A similar observation was also made for gap genes during short-germ segmentation in *Tribolium* (*Zhu et al., 2017*).

For prediction 2, in somitogenesis, there is an asymmetry in the wave pattern before stabilization. The transition from the high to low phase within one somite is shallower than the transition between those two phases from one somite to the other (i.e. posterior of one somite to anterior of the next), as detailed in *Shih et al., 2015*. A possible read-out is the downstream pattern of *Cadherin* controlling segment boundary formation, which presents a sawtooth profile (*McMillen et al., 2016*), consistent with the observed asymmetry in the wave pattern. It has been proposed that the main reason for the clock control was precisely to generate such periodic sawtooth pattern, and a SNIC bifurcation offers a plausible mechanism. In insects, it has been proposed that one of the roles of traveling waves of segmentation genes (e.g. *eve* genes shift) is indeed to provide segment polarity (*Rothschild et al., 2016*; *Clark, 2017*; *Clark and Akam, 2016*), which is consistent with this

proposal, although precise quantification of wave asymmetry has not been done in this context to our knowledge. In situs of gap genes in short germ insects show asymmetry between the anterior and posterior borders of the propagating gene expression waves as well (*Zhu et al., 2017*).

For prediction 3, the slowdown of oscillations during vertebrate segmentation appears quite variable between species. Mouse segmentation clock seems at first more consistent with a relatively sudden Hopf scenario. There is roughly a 2π phase shift between the posterior and the front, consistent with only a moderate change of the period/wavelength (*Lauschke et al., 2013*). Other species have several propagating waves within the PSM, seemingly more consistent with a broader period regime. In zebrafish, in vivo real-time imaging suggests a wavelength of propagating waves in the anterior PSM of roughly twice the segment length, meaning that the period of the clock at the front is at least half of the period in the tail bud (*Shih et al., 2015*). However, this measurement likely overestimates the wavelength and thus underestimates the period at the front since it is based on a peak to peak measurement between the last two waves. Thus, the period of the clock might further increase in the anterior, consistent with a period divergence and with measurements of *Giudicelli et al., 2007* on in situs, who proposed that the period indeed diverges (*Giudicelli et al., 2007*). Importantly, as said above, while the period increases the amplitude increases too *Shih et al., 2015*, which is more consistent with a SNIC bifurcation. In situs in snake PSM clearly show a decrease of the wavelength that seems to continuously match the pattern in the anterior, possibly more consistent with an infinite-period bifurcation as well (*Gomez et al., 2008*).

A direct experimental measure of the wavelength of the pattern for the last oscillations can be found with *Mesp2* (*Takahashi et al., 2000*; *Saga and Takeda, 2001*). The consensus on *Mesp2* is that it is expressed in the last few waves of oscillations, and that its pattern continuously regresses to reach exactly the wavelength of the anterior pattern. This is consistent with an infinite-period bifurcation. In *Oginuma et al., 2010* it is argued that Mesp helps setting the segment boundary in mouse by precisely reading the Notch wave, explaining both its biological role and why it is a convenient marker of the wavelength. We notice that the model used to explain the experiments in *Oginuma et al., 2010* is Julian Lewis' model from the appendix of *Palmeirim et al., 1997*, thus presenting an infinite-period divergence.

One cannot completely exclude from all those experimental observations a Hopf scenario where waves would not be very damped (or would be amplified for some reason) past the bifurcation, while the period of damped oscillations would become long until the saddle-node bifurcation is reached. A way to experimentally falsify this scenario is to induce a smoother transition and see if damped oscillations appear. Interestingly, both in vertebrates and in short germ insects, changes of *Wnt* signalling indeed give rise to smoother transitions from the dynamic to the static regime (suggesting that *Wnt* could be related to the control parameter *g*). In mouse, *beta-catenin* gain of function mutants give rise to considerably extended PSM toward the anterior (*Aulehla et al., 2008*), with up to five waves of oscillating *Lfng* (compared to only one in WT). Thus, this extended PSM qualitatively looks much more like a zebrafish or a snake PSM, with reinforcement of *Lfng* in situs from anterior to posterior (suggesting a refinement of the pattern and not damped oscillations, see Figure 4g,h of *Aulehla et al., 2008*). In the anterior PSM of such mutants, the wavelength of the oscillations decreases to match exactly the *Mesp2* wavelength (*Aulehla et al., 2008*, see e.g. *Figure 3*), and the very last waves move extremely slowly towards the anterior (Alexander Aulehla, private communication), consistent with a divergence of the time-scale. In *Tribolium*, *axin* RNAi similarly changes the wave pattern, where smoother and more spatially extended propagation of gap gene expressions are observed in those mutants, with reduced wavelengths compared to the WT (see Figure 4 of *Zhu et al., 2017*). Those phenotypes are exactly what is expected from an infinite-period bifurcation, again consistent with the SNIC scenario.

Lastly, damped oscillations would also be absent from Hopf scenarios where the bifurcation that destroys the limit cycle occurs at a value of the control parameter that is almost exactly the value at which the system becomes multistable. This happens for example when the supercritical Hopf bifurcation that kills the limit cycle happens at the same value of parameter *g* than the pitchfork bifurcation that generates bistability (*Figure 4—figure supplement 1F*). A similar phenomenon occurs in the model presented on *Figure 4—figure supplement 4*: the two simultaneous subcritical Hopf bifurcations that generate the bistability and the saddle-node of limit cycles that destroys the stable limit cycle happen in a very narrow range of *g* values. We cannot fully exclude those cases from the experimental data, but we notice they are intrinsically less generic since multiple bifurcations have to

occur at the same time, which happens only in our models because of specific parameter choices, symmetries in the equations or specific choices of non-linearity for the weights of the modules.

## Model predictions

The most straightforward prediction of the model proposed here is the presence of several global transcriptional modules between strongly interacting genes, directly controlling the smooth changes of developmental time-scale (in a similar way to the 'speed-gradient' model in *Zhu et al., 2017*). Many developmental genes are regulated by multiple 'shadow' enhancers (*Cannavò et al., 2016*). A smooth transition between different enhancers has even been observed for gap genes in *Drosophila* (*El-Sherif and Levine, 2016*). Global regulation of transcriptional modules could be biologically achieved through 'super enhancers' regulating many genes at the same time (*Hnisz et al., 2017*). A non-trivial prediction of our model is that the intrinsic time-scale of the system is a function of the relative balance of transcriptional activities of the modules. The transcriptional control described here naturally allows for infinite-period bifurcations, an implicit mechanism in several models of metazoan segmentation. This is to be contrasted with classical models of negative feedback oscillators such as the Goodwin model, where the time-scale is entirely controlled by degradation and is independent from transcription and translation rates (*Forger, 2011*), and with models of delayed oscillators, where the time-scale is essentially controlled by transcriptional delays (*Lewis, 2003*).

Our model is controlled by an external parameter $g$. The most natural hypothesis would be that $g$ corresponds to an actual morphogen gradient. As said above, *Wnt* is a natural candidate, but feedbacks clearly exist with *Caudal* in *Tribolium* (*Zhu et al., 2017*) and *FGF* in vertebrates. However, in the spirit of the initial wavefront proposal by Cooke and Zeeman, $g$ could also be in some context a temporal variable, for example an effective timer. Recent works on somitogenesis have suggested that the segmentation front could also be coupled to the slowing down of two oscillators (*Lauschke et al., 2013*), possibly one corresponding to *Notch/FGF* and the other to *Wnt* (*Sonnen et al., 2018*), so that the oscillation could feedback on itself to define $g$. One could also imagine that $g$ is related to more biophysical variables (density, elasticity) (*Hubaud et al., 2017*). It is important to point out that in our framework the nature of the bifurcation does *not* depend on the nature of $g$. While it might be difficult to experimentally disentangle feedbacks between the bifurcations and the control parameter from actual properties of the bifurcations themselves, our predictions on the nature of the bifurcations would not change.

An assumption of this two-module framework is that the same genes interact to control the system in *both* the dynamic and static regimes, giving rise to a smooth dynamical transition during development. This is consistent with what is observed for gap gene dynamics in short germ insects (*Zhu et al., 2017*). For vertebrate segmentation, we do not know yet mechanistically how both regimes are controlled, but the *Notch* signalling pathway is implicated in all steps of somitogenesis and in particular is known to gate information from the oscillatory to the segmented regime (*Oginuma et al., 2010*). An opposite view would be that the transition from dynamic to static regime is *de facto* sudden (even if it appears smooth for other reasons). Such scenario could be realized in different ways. For instance, different enhancers could regulate completely different sets of genes in the dynamic vs static regimes. The 'static' genes would then interact with the 'dynamic' genes only briefly during development, ensuring transmission of positional information between the static and dynamic regimes in a very localized region of time and space. In somitogenesis, as said above specific genes are indeed expressed at the so-called 'front' (such as *Mesp2*) and could act like gating processes transferring the information from the clock to an independent patterning system. In this case, we would be back to a sequential point of view where different regimes of development live in different regions of phase space, and the local bifurcation scenario would then be more plausible.

## Evolution and developmental plasticity

Evolutionary simulations for the evolution of patterning have favored a scenario based on Hopf and saddle-node bifurcations (*François et al., 2007*). Those simulations did not include multiple enhancers like here, and all transcriptional regulations had essentially to evolve from scratch, possibly suggesting a 'Ur-segmentation clock' based on Hopf and saddle-node bifurcations. This scenario is not excluded by our model: in particular there would be no difference between Hopf and SNIC bifurcations under the control of steep gradients of $g$, which shortcut the region where only one

fixed point is present. So Hopf/saddle node bifurcations under control of a steep gradient are likely the easiest solution found by (simulated) evolution. However, adding a more complex, enhancer-based evolutionary grammar might allow for more combinatorial use of dynamical attractors, and associated robustness to external perturbations (such as the shape of the $g$ gradient). A SNIC bifurcation might then plausibly evolve from the modularization (*West-Eberhard, 2003*) of a system based on Hopf and saddle-node bifurcations (both in vivo and in silico). First, since the SNIC bifurcation is the generic scenario that we observe in our framework, it should be easy to discover by evolution once a multi-enhancer system combining an oscillator and a bistable system has evolved. Second, SNIC-controlled segmentation is plastic in the sense that changes of dynamics of the control parameter would change the transient phenotypes (such as the number of oscillating waves) but would still generate a pattern. Modularizations leading to developmental plasticity has been suggested to be an important engine for evo-devo (*West-Eberhard, 2003*), since it allows for intra-specific variability without impinging on the most important phenotypes (here, segmentation).

Importantly, such plasticity is indeed observed experimentally both for short germ and vertebrate segmentation. For instance, in *Tribolium* one can considerably modify *Caudal* gradient dynamics and still see proper patterning (*Zhu et al., 2017*). It could thus explain how and why there is so much quantitative variability in segmentation mechanisms, such as short vs intermediate germ band segmentation (as suggested in *Zhu et al., 2017*). In somitogenesis, there is a lot of interspecies quantitative variability in the numbers of waves and rescaled periods (*Gomez et al., 2008*) while the qualitative dynamics itself appears very conserved see for example (*Krol et al., 2011*). In other words, a two-module mechanism makes the dynamics both more robust – a generic bifurcation scenario gives precise phase encoding – and more evolvable – one can vary many features of the system (e.g. basins of attractions, dynamics of the control parameter) and still get proper patterning, again a hallmark of developmental plasticity.

In brief, we have discussed a two-module based model of the smooth transition in development from a dynamical regime to a static one. This model explains time-scale divergence, as well as robustness to changes of morphogen dynamics (*Zhu et al., 2017*) and to noise. It provides a possible explanation for smooth robust transitions in metazoan segmentation, with a non-trivial (global) bifurcation. Further experimental and theoretical studies are required to assess the importance of smooth transitions for encoding dynamic information into spatial patterns of genetic expressions.

## Acknowledgements

We thank members of the François and El-Sherif groups for insightful discussions, as well as referees for their insightful comments. This work was supported by a CREATE: Complex Dynamics fellowship and a FRQNT B2 scholarship to L Jutras-Dubé, a DFG grant EL 870/2–1 to E El-Sherif, a NSERC Discovery Grant and a Simons Investigator in Mathematical Modelling of Living Systems to P François.

## Additional information

### Funding

| Funder | Grant reference number | Author |
|---|---|---|
| Simons Foundation | MMLS | Laurent Jutras-Dubé<br>Paul François |
| Natural Sciences and Engineering Research Council of Canada | CREATE in Complex Dynamics | Laurent Jutras-Dubé |
| Fonds de Recherche du Québec - Nature et Technologies | B2 | Laurent Jutras-Dubé |
| Deutsche Forschungsgemeinschaft | EL 870/2-1 | Ezzat El-Sherif |
| Natural Sciences and Engineering Research Council of Canada | Discovery Grant | Paul François |

The funders had no role in study design, data collection and interpretation, or the decision to submit the work for publication.

### Author contributions
Laurent Jutras-Dubé, Conceptualization, Data curation, Formal Analysis, Investigation, Methodology, Software, Validation, Visualization, Writing – original draft, Writing – review & editing; Ezzat El-Sherif, Conceptualization, Investigation, Writing - original draft, Writing - review and editing; Paul François, Conceptualization, Resources, Formal analysis, Supervision, Funding acquisition, Investigation, Methodology, Writing - original draft, Project administration, Writing - review and editing

### Author ORCIDs
Laurent Jutras-Dubé  https://orcid.org/0000-0003-4323-2840
Ezzat El-Sherif  https://orcid.org/0000-0003-1738-8139
Paul François  https://orcid.org/0000-0002-2223-839X

### Decision letter and Author response
Decision letter https://doi.org/10.7554/eLife.55778.sa1
Author response https://doi.org/10.7554/eLife.55778.sa2

## Additional files
### Supplementary files
• Transparent reporting form

### Data availability
https://github.com/laurentjutrasdube/Dual-Regime_Geometry_for_Embryonic_Patterning (copy archived at https://github.com/elifesciences-publications/Dual-Regime_Geometry_for_Embryonic_Patterning).

The following dataset was generated:

| Author(s) | Year | Dataset title | Dataset URL | Database and Identifier |
|---|---|---|---|---|
| Jutras-Dubé L, Sherif EE, François P | 2020 | Geometric models for robust encoding of dynamical information into embryonic patterns | https://github.com/laurentjutrasdube/Dual-Regime_Geometry_for_Embryonic_Patterning | Github, laurentjutrasdube/ Dual-Regime_ Geometry_for_ Embryonic_Patterning |

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

## Appendix 1

### 1 A two-enhancer model reproduces dynamical features of tribolium segmentation

In *Zhu et al., 2017*, two of us proposed a model of Tribolium segmentation relying on the interplay of two sets of enhancers. In short, two sets of enhancers (static $S(P)$, dynamic $D(P)$) were used, where the role of parameter g is played by morphogen *Caudal* (*cad*) (*El-Sherif et al., 2012*; *Figure 1—figure supplement 2*). $S(P)$ encodes a multistable system and $D(P)$ a sequential cascade of genetic expression of gap genes (*hb*, *Kr*, *mlpt*, *gt*). This system was found to implement a 'speed gradient' model, where the speed of traveling waves of gap genes from posterior to anterior depended on the level of *cad* concentration (*Figure 1—figure supplement 2B–D*). This led to robust patterning of the embryo (*Figure 1—figure supplement 2E*) but the mathematical origin of the speed gradient was not explained.

To better understand the underlying dynamics of the system, we consider the time courses of multiple cells at different positions and thus with different final fates. *Figure 1—figure supplement 2F* shows the projection of the cells' dynamics on a 2D plane corresponding to the first two genes expressed in the cascade (*Kr* and *hb*), as well as a typical flow for different values of *cad* while keeping the other genes (*mlpt*, *gt* and *X*) at zero. Importantly, for $cad = 0.13$, we see the appearance of a new fixed point (green disk on *Figure 1—figure supplement 2F*).

We make four observations:

- The flow of the system is canalized. The trajectories of the cells stay very close to one another in phase space.
- As *cad* is lowered, the new fixed point appears very close to the common trajectory of all cells (*Figure 1—figure supplement 2F*, top row), and clearly separates the trajectories of cells ending up at different fates (*Figure 1—figure supplement 2F*, bottom row).
- When *cad* further decreases, the new fixed point moves in the high *hb*, low *Kr* region, corresponding to the eventual fate of Cell 1.
- When the new fixed point appears, the flow of cells past this fixed point is slowed down (*Figure 1—figure supplement 2G*).

These four observations offer a concise explanation to the 'speed gradient' model: as the system gets closer to the bifurcation happening at $cad = 0.13$, the system is slowing down because of the future fixed points appearing on the trajectory. Intuitively, this is due to the fact that a fixed point corresponds to a frozen state, and thus to an infinite time-scale (static). When *cad* varies, the system has to interplay between a non-zero time-scale (dynamics) and such infinite time-scale, and it thus makes sense a priori that in between, the time-scale of the system diverges. This mechanism is close in principle to the critical timing proposed in *Tufcea and François, 2015*.

### 2 List of the functions used for the dynamics of each model

#### 2.1 Gene network models

In the gene network models, biochemical interactions between genes are modeled explicitly. Ordinary differential equations (ODEs) represent the dynamics of the concentration of the proteins that are encoded by the genes in the network. The deterministic part of the dynamics is composed of a protein production term and a protein degradation term. The production rate of a given protein can be altered by the interactions between the genes. Hill functions are use to model repression and activation of the production of a given protein by the genes. When multiple genes affect the concentration of a protein, the Hill functions corresponding to each interaction are multiplied. In the simulations, we set to one the maximal production rate of all proteins. Similarly, we set the degradation rate of all proteins to 1. In *Equation 1* of the main text, $C(P)$ encodes the degradation term, and $\Theta_S(g)\,S(P) + \Theta_D(g)\,D(P)$ represents the production term.

#### 2.1.1 3-gene models

The proteins associated to the three genes are named arbitrarily $A$, $B$ and $C$:

$$P = \begin{bmatrix} A \\ B \\ C \end{bmatrix} \quad C(P) = \begin{bmatrix} -A \\ -B \\ -C \end{bmatrix} \quad D(P) = \begin{bmatrix} \frac{1}{1+(B/K_D^{B \dashv A})^5} \\ \frac{1}{1+(C/K_D^{C \dashv B})^5} \\ \frac{1}{1+(A/K_D^{A \dashv C})^5} \end{bmatrix} \quad (P) = \begin{bmatrix} \frac{1}{1+(B/K_S^{B \dashv A})^5} & \frac{1}{1+(C/K_S^{C \dashv A})^5} \\ \frac{1}{1+(C/K_S^{C \dashv B})^5} & \frac{1}{1+(A/K_S^{A \dashv B})^5} \\ \frac{1}{1+(A/K_S^{A \dashv C})^5} & \frac{1}{1+(B/K_S^{B \dashv C})^5} \end{bmatrix} \tag{1}$$

*Appendix 1—table 1* lists the values of the parameters used in the repression interactions of all versions of the 3-gene models: the symmetric version used to generate the results of *Figure 2*, *Figure 2—figure supplements 1* and *2*, *Figure 3*, *Figure 3—figure supplement 1* and *Figure 7—figure supplement 2*, the version with a weak asymmetry used in *Figure 5*, the version with a strong asymmetry used in *Figure 5—figure supplement 1* and the version with a randomized asymmetry used in *Figure 5—figure supplement 2*. In the latter version, we randomly picked the values of the repression interactions of the static term $S(P)$ from a Gaussian distribution with mean 0.4 and standard deviation 0.04. *Appendix 1—table 2* lists the weights $\Theta_D(g)$ and $\Theta_S(g)$ used for all 3-gene models: Models 1 and 2 used to generate the results of *Figure 2*, *Figure 2—figure supplement 1*, *Figure 3*, *Figure 5*, *Figure 5—figure supplements 1* and *2* and *Figure 7—figure supplement 2*, as well as Models 1 and 2 with Hill functions for the weights $\Theta_D(g)$ and $\Theta_S(g)$, used in *Figure 2—figure supplement 1*, *Figure 3—figure supplement 1* and *Figure 7—figure supplement 2*.

**Appendix 1—table 1.** Parameter values for the repression interactions of the 3-gene models.

| Model version | $K_D^{B \dashv A}$ | $K_D^{C \dashv B}$ | $K_D^{A \dashv C}$ | $K_S^{B \dashv A}$ | $K_S^{C \dashv A}$ | $K_S^{C \dashv B}$ | $K_S^{A \dashv B}$ | $K_S^{A \dashv C}$ | $K_S^{B \dashv C}$ |
|---|---|---|---|---|---|---|---|---|---|
| Symmetric | 0.4 | 0.4 | 0.4 | 0.4 | 0.4 | 0.4 | 0.4 | 0.4 | 0.4 |
| Weak asymmetry | 0.4 | 0.4 | 0.4 | 0.36 | 0.36 | 0.4 | 0.4 | 0.4 | 0.4 |
| Strong asymmetry | 0.4 | 0.4 | 0.4 | 0.32 | 0.32 | 0.36 | 0.36 | 0.4 | 0.4 |
| Randomized asymmetry | 0.4 | 0.4 | 0.4 | 0.3825 | 0.3560 | 0.4334 | 0.4102 | 0.3802 | 0.4038 |

**Appendix 1—table 2.** Weights of the dynamic and static terms of the 3-gene models.

| Weights | Model 1 | Model 2 | Model 1 with hill functions | Model 2 with hill functions |
|---|---|---|---|---|
| $\Theta_D(g)$ | $g^2$ | $g$ | $\frac{(g/0.4)^5}{1+(g/0.4)^5}$ | $\frac{(g/0.4)^5}{1+(g/0.4)^5}$ |
| $\Theta_S(g)$ | $(1-g)^2$ | $1-g$ | $\frac{1}{1+(g/0.6)^5}$ | $\frac{1}{1+(g/0.4)^5}$ |

## 2.1.2 Model of Tribolium segmentation

In the model of *Figure 1—figure supplement 2*, the interactions between *hunchback* (*hb*), *Krüppel* (*Kr*), *mille-pattes* (*mlpt*), *giant* (*gt*) and an unidentified gene *X* are modeled (see the supplement of *Zhu et al., 2017*). Note that the role of parameter *g* is played by *caudal* (*cad*) in the model for Tribolium segmentation.

$$P = \begin{bmatrix} hb \\ Kr \\ mlpt \\ gt \\ X \end{bmatrix} \quad C(P) = \begin{bmatrix} -hb \\ -Kr \\ -mlpt \\ -gt \\ -X \end{bmatrix} \quad D(P) = \begin{bmatrix} \frac{(hb/0.2)^5}{1+(hb/0.2)^5} \frac{1}{1+(Kr/0.12)^5} \\ \frac{(hb/0.4)^5}{1+(hb/0.4)^5} \frac{1}{1+(mlpt/0.25)^5} \frac{1}{1+(gt/0.01)^5} \\ \frac{(Kr/0.4)^5}{1+(Kr/0.4)^5} \frac{1}{1+(gt/0.3)^5} \\ \frac{(mlpt/0.4)^5}{1+(mlpt/0.4)^5} \frac{1}{1+(X/0.08)^5} \\ \frac{(gt/0.4)^5}{1+(gt/0.4)^5} \end{bmatrix} \tag{2}$$

$$S(P) = \begin{bmatrix} \dfrac{(hb/0.4)^5}{1+(hb/0.4)^5}\dfrac{1}{1+(Kr/0.4)^5} \\[2ex] \dfrac{(Kr/0.4)^5}{1+(Kr/0.4)^5}\dfrac{1}{1+(hb/0.01)^5} \\[2ex] \dfrac{(mlpt/0.4)^5}{1+(mlpt/0.4)^5} \\[2ex] \dfrac{(gt/0.4)^5}{1+(gt/0.4)^5} \\[2ex] \dfrac{(X/0.4)^5}{1+(X/0.4)^5} \end{bmatrix} \qquad \Theta_D(cad) = 3\,\frac{cad}{1+cad} \qquad \Theta_S(cad) = \frac{1}{1+cad} \tag{3}$$

## 2.2 Gene-free models

In the gene-free model, ODEs encode flows in an abstract 2D phase space. The two geometric variables are named arbitrarily $y$ and $z$:

$$P = \begin{bmatrix} y \\ z \end{bmatrix} \qquad C(P) = \begin{bmatrix} 0 \\ 0 \end{bmatrix} \qquad D(P) = \begin{bmatrix} y\left(1-\sqrt{y^2+z^2}\right)-z \\ z\left(1-\sqrt{y^2+z^2}\right)+y \end{bmatrix}$$

$$S(P) = \begin{bmatrix} (y_0-y)\,(y_1-y)\,(y_2-y) \\ -z \end{bmatrix} \tag{4}$$

where parameter $y_0$ (resp. $y_1$ and $y_2$) controls the position of the unstable fixed point (resp. the stable fixed points) of the static term along the $y$ axis. Parameter $y_0$ is set to 0 in the symmetric version of the model used in *Figure 4*, *Figure 4—figure supplements 1*, *2*, *3*, *4* and *5*, *Figure 4—videos 1*, *2*, *3*, *4*, *Figure 6*, *Figure 7*, *Figure 7—figure supplement 1* and *Figure 7—video 1*, as well as in the asymmetric versions of *Figure 6—figure supplement 1* and *Figure 7—video 1*. In *Figure 6*, parameter $y_0$ is set to 0.05 and 0.1 to model different levels of asymmetry in the basins of attraction. In *Figure 7—figure supplement 1*, parameter $y_0$ is set to 0.02 for Model 1 and 0.05 for Model two to obtain a similar level of asymmetry in the final pattern generated by the two models. We set $y_1 = -1$ and $y_2 = 1$ for all versions of the gene-free models, except for the asymmetric versions used to generate the results of *Figure 6—figure supplement 1* and *Figure 7—video 1*. In the former, the stable fixed points of the static module are placed outside the region delimited by the limit cycle of the dynamic module by setting $y_1 = -2$ and $y_2 = 2.5$, while in the latter, we set $y_1 = -1.75$ and $y_2 = 1$.

To obtain a supercritical Hopf bifurcation with the gene-free model, we followed a similar approach than for the 3-gene model. We reasoned that the sum of the weights of the dynamic and static modules should become smaller than a degradation-like term for values of $g$ around 0.5. For this reason, an 'intermediate term' $I(P) = [-z \quad -y]^T$ is introduced in the ODE. The intermediate term is weighted by the function $\Theta_I(g)$. *Equation 1* of the main text thus becomes:

$$\dot{P} = \Theta_D(g)\,D(P) + \Theta_I(g)\,I(P) + \Theta_S(g)\,S(P) + \eta(g,P) \tag{5}$$

Recall that in a given cell, only the dynamic module should be present at the beginning of the simulation, when $g = 1$. Similarly, only the static module should be present at the end of the simulation, when $g = 0$. Therefore, we set the weight of the intermediate module equal to $g\,(1-g)$, which is zero at both $g = 1$ and $g = 0$. Since this weight is of the order two in $g$, we make the weights of the dynamic and static modules of the order three in $g$ to ensure that they become smaller than the weight of the intermediate term for $g$ around 0.5. To obtain subcritical Hopf bifurcations with the gene-free model, we used a slightly different dynamic module:

$$D(P) = \begin{bmatrix} y\,\sqrt{y^2+z^2}\left(1-\sqrt{y^2+z^2}\right)-z \\ z\,\sqrt{y^2+z^2}\left(1-\sqrt{y^2+z^2}\right)+y \end{bmatrix} \tag{6}$$

*Appendix 1—table 3* lists the weights used for all gene-free models: Model one used to generate the results of *Figure 4—figure supplements 1*, *2* and *5*, *Figure 4—video 1*, *Figure 6*, *Figure 6—figure supplement 1*, *Figure 7—figure supplements 1* and *2*, and *Figure 7—video 1*, Model two used to generate the results of *Figure 4*, *Figure 4—figure supplements 1* and

5, *Figure 4—video 2*, *Figure 6*, *Figure 6—figure supplement 1*, *Figure 7*, *Figure 7—figure supplements 1* and *2*, and *Figure 7—video 1*, Model three used to generate the results of *Figure 4—figure supplements 3* and *5*, *Figure 4—video 3* and *Figure 7—figure supplement 2*, and Model four used to generate the results of *Figure 4—figure supplements 4* and *5*, *Figure 4—video 4* and *Figure 7—figure supplement 2*.

**Appendix 1—table 3.** Weights of the dynamic, static and intermediate terms of the gene-free models.

| Weights | Models 1 and 3 | Models 2 and 4 |
|---|---|---|
| $\Theta_D(g)$ | $g^3$ | $g$ |
| $\Theta_S(g)$ | $(1-g)^3$ | $1-g$ |
| $\Theta_I(g)$ | $g > (1-g)$ | $0$ |

## 2.3 Infinite-period scenarios of *Figure 1* and *Figure 7*

The infinite-period scenario of *Figure 1B–F* is a simplified version of the model of the appendix of *Palmeirim et al., 1997*. The dynamics of the phase of the oscillators are modeled directly using the following ODE:

$$\dot{\phi} = \omega(g) = \frac{\pi}{2} g^2 \tag{7}$$

The infinite-period scenario of *Figure 7A–E* is the 1D model of coupled oscillators from *Gillespie, 2001*. In brief, the dynamics of the phase of the oscillators are described by the following ODE:

$$\dot{\phi}(x,t) = \omega(x,t) + \frac{\epsilon}{2a^2}\left(\sin[\phi(x-a,t-\tau) - \phi(x,t)] + \sin[\phi(x+a,t-\tau) - \phi(x,t)]\right) \tag{8}$$

where $\epsilon$ represents the coupling strength between a cell and its two nearest neighbors, $a$ is the average cell diameter (cd), and $\tau$ is the time delay in the coupling. The spatio-temporal profile of the frequency of the oscillators $\omega(x,t)$ is given by the following formula:

$$\omega(x,t) = \omega_\infty\left(1 - e^{-(x-vt)/\sigma}\right) \tag{9}$$

where $\omega_\infty$ represents the characteristic intrinsic frequency of the oscillators, $v$ is the speed at which the spatial frequency profile moves along the posterior direction, and $\sigma$ controls the spatial steepness of the frequency profile. *Appendix 1—table 4* lists the parameter values used to generate the results of *Figure 7A–E*. See *Gillespie, 2001* for more details.

**Appendix 1—table 4.** Parameter values for the ODE of the phase oscillators in the infinite-period scenario of *Figure 7*.

| $\epsilon$ [cd²/min] | $a$ [cd] | $\tau$ [min] | $\omega_\infty$ [min⁻¹] | $v$ [cd/min] | $\sigma$ [cd] |
|---|---|---|---|---|---|
| 0.07 | 1 | 0 | 0.3886 | 0.255 | 36 |

## 2. 4 Hopf scenario of *Figure 1*

The Hopf scenario of *Figure 1G–K* is the cell-autonomous model evolved in silico in *François et al., 2007*. The model describes the dynamics of two proteins, the effector protein $E$ and the repressor protein $R$, under the control of morphogen $g$ via ODEs with time delays:

$$\dot{E} = \left(\max\left[\frac{E^{n_1}}{E^{n_1} + E_E^{n_1}}, \frac{g^{n_2}}{g^{n_2} + g_E^{n_2}}\right]\frac{S_E}{1 + (R/R_E)^{n_3}}\right)_{t-\tau_E} - \delta_E E \tag{10}$$

$$\dot{R} = \left(\frac{g^{n_4}}{g^{n_4} + g_R^{n_4}}\frac{S_R}{1 + (R/R_R)^{n_5}}\right)_{t-\tau_R} - \delta_R R \tag{11}$$

The subscript of a closed parenthesis indicates the time at which the expression inside the parenthesis is evaluated. If no such parenthesis with a subscript is present in a given expression, this expression is evaluated at time $t$. The values of all parameters are given in *Appendix 1—tables 5* and *6*

**Appendix 1—table 5.** Parameter values for the ODE of the effector protein $E$ in the Hopf scenario of *Figure 1*.

| $S_E$ | $R_E$ | $g_E$ | $E_E$ | $\tau_E$ | $\delta_E$ | $n_1$ | $n_2$ | $n_3$ |
|---|---|---|---|---|---|---|---|---|
| 0.7176 | 0.4942 | 0.0678 | 0.3213 | 0.48 | 0.8538 | 3 | 4.3549 | 4.5321 |

**Appendix 1—table 6.** Parameter values for the ODE of the repressor protein $R$ in the Hopf scenario of *Figure 1*.

| $S_R$ | $R_R$ | $g_R$ | $\tau_R$ | $\delta_R$ | $n_4$ | $n_5$ |
|---|---|---|---|---|---|---|
| 0.9422 | 0.1156 | 0.5047 | 3.92 | 0.9759 | 3.2136 | 4.522 |

## 2.5 Van der Pol oscillator of *Figure 7*

The schematic of an 'asymmetric wave' profile shown on *Figure 7D* is generated with the following ODEs describing a Van der Pol oscillator:

$$\dot{y} = \mu \left( y - \frac{y^3}{3} - z \right) \tag{12}$$

$$\dot{z} = \frac{y}{\mu} \tag{13}$$

where parameter μ is set to 2.5. The schematic of a "symmetric wave" profile shown on *Figure 7D* is a sine function.

## 3 Spatio-temporal profile of the control parameter for each model

For all models except the model for Tribolium segmentation and the infinite-period scenario of *Figure 7A–E*, the following function is used to describe the spatio-temporal profile of the input $g$, which is treated either as the concentration of a morphogen in the gene network models, or as an abstract control parameter in the gene-free models:

$$g(x,t) = H(x - vt) = \min\left[ e^{s\,(x - vt + x_{\mathrm{osc}})}, 1 \right] \tag{14}$$

where parameter $s$ controls the steepness of the gradient and $v$ represents the speed at which the gradient moves along the antero-posterior axis. Parameter $x_{\mathrm{osc}}$ allows to generate a few oscillations inside the first simulated cell before $g$ starts decreasing. Note that the position vector $x$ is normalized in all our simulations, such that positions are constrained from 0 to 1. *Appendix 1—table 7* lists the values of the parameters used for the gradients of all models (except the model for Tribolium segmentation): the gradients of the infinite-period scenario and of the Hopf scenario used to generate the results of *Figure 1B–F* and *Figure 1G–K*, respectively, the shallow gradient used in the 3-gene models of *Figure 2*, *Figure 2—figure supplement 1* and *2*, *Figure 3*, *Figure 3—figure supplement 1*, *Figure 5*, *Figure 5–figure supplements 1* and *2*, and *Figure 7—figure supplement 2*, the steep gradient used in the 3-gene models of *Figure 5*, *Figure 5–figure supplements 1* and *2* and , and the gradients used in the gene-free models of *Figure 4*, *Figure 4–figure supplements*

*2*, *3*, *4* and *5*, *Figure 6*, *Figure 6—figure supplement 1*, *Appendix 1—table 7*, *Figure 7—figure supplements 1* and *2* and *Figure 7—video 1*.

**Appendix 1—table 7.** Parameter values for the spatio-temporal profile of input *g*.

| Model | S | V | $x_{\mathrm{osc}}$ |
|---|---|---|---|
| Infinite-period scenario of *Figure 1* | 0.5 | 0.08 | 0.2 |
| Hopf scenario of *Figure 1* | 0.5 | 3 | 0 |
| 3-gene models, shallow gradient | 1 | 0.05 | 0.2 |
| 3-gene models, steep gradient | 2.5 | 0.05 | 0.2 |
| Gene-free models (*Figure 4*, its supplements and *Figure 7—figure supplement 2*) | 0.5 | 0.035 | 0.2 |
| Gene-free models (*Figure 6* and its supplement) | 1 | 0.036 | 0 |
| Gene-free models (*Figure 7* and *Figure 7—figure supplement 1*) | 6 | 0.0042 | 0 |
| Gene-free models (*Figure 7—video 1*) | 0.9 | 0.04 | 0 |

In the model for *Tribolium* segmentation, the role of input *g* is played by the maternal gene *cad*. The dynamics of *cad* is modelled with a Hill function:

$$cad(x,t) = \frac{(x/x^*(t))^{n(t)}}{1 + (x/x^*(t))^{n(t)}} \tag{15}$$

where the time dependencies of parameters $x^*(t)$ and $n(t)$ encode respectively the regression of the morphogen gradient along the antero-posterior axis, and the gradual increase in the steepness of the morphogen gradient:

$$x^*(t) = \max[0.4 , \, 0.4 + 0.2 \, (t-2)] \quad ; \quad n(t) = \max[4 , \, 4 \, e^{(t-2)}] \tag{16}$$

## 4 Integration schemes

### 4. 1 Euler algorithm for deterministic simulations

*Equation 1* of the main text can be integrated via the Euler algorithm to obtain a time series representing the deterministic dynamics of vector *P*:

$$P(t+dt) = P(t) + \big( \Theta_D(g(t)) \, D(P(t)) + \Theta_S(g(t)) \, S(P(t)) + C(P(t)) \big) \, dt \tag{17}$$

The Euler algorithm, which is equivalent to approximating the temporal derivative of *P* by a first-order finite difference, was used to perform deterministic simulations of all versions of the 3-gene models (*Figure 2*, *Figure 2—figure supplements 1* and *2*, *Figure 5*, *Figure 5—figure supplements 1* and *2* and *Figure 7—figure supplement 2*). A similar version of this algorithm that includes the intermediate term was used for deterministic simulations of the gene-free models (*Figure 4*, *Figure 4—figure supplements 2*, *3* and *4*, *Figure 6*, *Figure 6—figure supplement 1*, *Figure 7*, *Figure 7—figure supplements 1* and *2* and *Figure 7—video 1*). The Euler algorithm was also used to perform simulations of the infinite-period and Hopf scenarios (*Figure 1* and *Figure 7*). On the other hand, deterministic simulations of the model for *Tribolium* segmentation were carried out via the lsoda integrator from the scipy library in Python (*Figure 1—figure supplement 2*).

### 4.2 Langevin equation for stochastic simulations of the 3-gene models

The stochastic nature of chemical reactions, due at least partly to the finite number of molecules involved in these reactions, introduces fluctuations in protein concentrations in single cells. To generate the results of *Figure 3* and *Figure 3—figure supplement 1*, noise was introduced in the 3-gene models in a chemically realistic and mathematically rigorous way by following the method of *Gillespie, 2001*. In the generic formulation of the present problem, there are $N$ molecular species $S_i$, $i = 1, ..., N$, that can interact through $M$ different reactions $R_j$, $j = 1, ..., M$. Let $X_i(t)$ represent the

number of $S_i$ molecules at time $t$. Then, the vector $X(t) \equiv [X_i(t) \quad ... \quad X_N(t)]$ represents the state of the whole system of $N$ molecules at time $t$. For each reaction $R_j$, a propensity function $a_j$ is defined such that if the system is in state $X$ at time $t$, then $a_j(X)\,dt$ is the probability that one $R_j$ reaction will occur in the next infinitesimal time interval $dt$, i.e. between $t$ and $t+dt$. For each reaction $R_j$, a state-change vector $\nu_j$ is defined such that its $i$th component $\nu_{ji}$ represents the change in the number of $S_i$ molecules produced by one $R_j$ reaction. Once the $M$ propensity functions and state-change vectors are defined, the time evolution of the state vector $X(t)$ is found via the $N$ deterministic reaction rate equations:

$$\dot{X}_i(t) = \sum_{j=1}^{M} \nu_{ji}\, a_j(X(t)) \quad \text{for } i = 1,...,N \tag{18}$$

The numerical integration of these rate equations can be performed via the Euler algorithm:

$$X_i(t+dt) = X_i(t) + \sum_{j=1}^{M} \nu_{ji}\, a_j(X(t))\, dt \quad \text{for } i = 1,...,N \tag{19}$$

The stochastic form of this simulation algorithm is given by the chemical Langevin equation:

$$X_i(t+dt) = X_i(t) + \sum_{j=1}^{M} \nu_{ji}\, a_j(X(t))\, dt + \sum_{j=1}^{M} N_j(t)\, \nu_{ji}\, \sqrt{a_j(X(t))\, dt} \quad \text{for } i = 1,...,N \tag{20}$$

where $N_1(t),...,N_M(t)$ are $M$ independent Gaussian random variables with mean and variance equal to 0 and 1, respectively, and that are not correlated in time. In the 3-gene models, the role of vector $X$ is played by $P$. Note that re-scaling the numbers of proteins $X_i$ by constant factors corresponds to multiplying both sides of *Equations 18, 19, 20* by that constant factor (as long as the state-change vectors $\nu_j$ are also re-scaled). Therefore, *Equations 18, 19, 20* are still valid when simulating protein concentrations scaled from 0 to 1 instead of absolute numbers of proteins. Furthermore, the reactions of the 3-gene models are encoded in the protein production and degradation terms. The propensities of the protein production and degradation terms are respectively $\Theta_D(g)\,D(P) + \Theta_S(g)\,S(P)$ and $P$. *Equation 19* thus becomes *Equation 17*, and *Equation 20* can be re-written as the following expression:

$$\begin{aligned} P_i(t+dt) &= P_i(t) + \big(\Theta_D(g(t))\,D_i(P(t)) + \Theta_S(g(t))\,S_i(P(t)) - P_i(t)\big)\,dt \quad i = 1,2,3 \\ &+ \Big(N_i^{\mathrm{prod}}(t)\sqrt{\Theta_D(g(t))D_i(P(t)) + \Theta_S(g(t))S_i(P(t))} - N_i^{\mathrm{deg}}(t)\sqrt{P_i(t)}\Big)\sqrt{dt} \end{aligned} \tag{21}$$

where $N^{\mathrm{prod}}(t) = [N_1^{\mathrm{prod}}(t), N_2^{\mathrm{prod}}(t), N_3^{\mathrm{prod}}(t)]$ and $N^{\mathrm{deg}}(t) = [N_1^{\mathrm{deg}}(t), N_2^{\mathrm{deg}}(t), N_3^{\mathrm{deg}}(t)]$ are 2 vectors, each containing 3 independent Gaussian random variables with mean 0 and variance 1. This equation can be simplified by leveraging the fact that the sum of Gaussian random variables with mean 0 and different variances is equal to a single Gaussian random variable with mean 0 and a variance equal to the sum of the variances:

$$\begin{aligned} P_i(t+dt) &= P_i(t) + \big(\Theta_D(g(t))\,D_i(P(t)) + \Theta_S(g(t))\,S_i(P(t)) - P_i(t)\big)\,dt \quad i = 1,2,3 \\ &+ \Big(N_i(t)\sqrt{\Theta_D(g(t))\,D_i(P(t)) + \Theta_S(g(t))\,S_i(P(t)) + P_i(t)}\Big)\sqrt{dt} \end{aligned} \tag{22}$$

where $N(t) = [N_1(t), N_2(t), N_3(t)]$ is a vector containing 3 independent Gaussian random variables with mean 0 and variance 1. Note that a different independent random variable is used for each protein, since the production term of each protein is due to a different combination of repression interactions. To control the level of noise, a parameter $\Omega$ is introduced in the previous equation such that increasing $\Omega$ decreases the level of noise:

$$\begin{aligned} P_i(t+dt) &= P_i(t) + \big(\Theta_D(g(t))\,D_i(P(t)) + \Theta_S(g(t))\,S_i(P(t)) - P_i(t)\big)\,dt \quad i = 1,2,3 \\ &+ \Big(\frac{N_i(t)}{\sqrt{\Omega}}\sqrt{\Theta_D(g(t))\,D_i(P(t)) + \Theta_S(g(t))\,S_i(P(t)) + P_i(t)}\Big)\sqrt{dt} \end{aligned} \tag{23}$$

Since noise arises at least partly from the stochastic nature of single reactions between a finite

number of proteins, increasing the concentration of proteins is expected to buffer the intrinsic chemical noise. Therefore, the noise level is expected to decrease as the protein concentration is increased. The following mathematical derivation shows that parameter $\Omega$ can be interpreted as the typical concentration of proteins in the system, such that increasing the protein concentration corresponds to increasing the value of parameter $\Omega$. First, let's take a look at the stochastic integration algorithm for protein $A$ and write explicitly the maximal production rate $\rho_A$ and the degradation rate $\delta_A$:

$$
\begin{aligned}
A^+ &= A + \left( \rho_A \left( \Theta_D(g) \frac{1}{1+(B/K_D^{B \dashv A})^5} + \Theta_S(g) \frac{1}{1+(B/K_S^{B \dashv A})^5} \frac{1}{1+(C/K_S^{C \dashv A})^5} \right) - \delta_A A \right) dt \\
&+ \frac{N_1}{\sqrt{\Omega}} \sqrt{\rho_A \left( \Theta_D(g) \frac{1}{1+(B/K_D^{B \dashv A})^5} + \Theta_S(g) \frac{1}{1+(B/K_S^{B \dashv A})^5} \frac{1}{1+(C/K_S^{C \dashv A})^5} \right) + \delta_A A} \ \sqrt{dt}
\end{aligned}
\tag{24}
$$

where a + superscript on a protein concentration indicates that this variable is evaluated at time $t + dt$ and the absence of a superscript on a variable indicates that it is evaluated at time $t$. Multiplying both sides of the equation by $\Omega$ leads to the following expression:

$$
\begin{aligned}
\Omega A^+ &= \Omega A + \left( \Omega \rho_A \left( \Theta_D(g) \frac{1}{1+(B/K_D^{B \dashv A})^5} + \Theta_S(g) \frac{1}{1+(B/K_S^{B \dashv A})^5} \frac{1}{1+(C/K_S^{C \dashv A})^5} \right) - \Omega \delta_A A \right) dt \\
&+ N_1 \sqrt{\Omega \rho_A \left( \Theta_D(g) \frac{1}{1+(B/K_D^{B \dashv A})^5} + \Theta_S(g) \frac{1}{1+(B/K_S^{B \dashv A})^5} \frac{1}{1+(C/K_S^{C \dashv A})^5} \right) + \Omega \delta_A A} \sqrt{dt}
\end{aligned}
\tag{25}
$$

Now, let's re-scale all quantities that have the units of protein concentration by a factor of $\Omega$. To achieve this, we define the re-scaled variables $A^* = \Omega A$, $B^* = \Omega B$ and $C^* = \Omega C$, as well as re-scaled parameters $\rho_{A^*} = \Omega \rho_A$, $K_D^{B^* \dashv A^*} = \Omega K_D^{B \dashv A}$, $K_S^{B^* \dashv A^*} = \Omega K_D^{B \dashv A}$ and $K_S^{C^* \dashv A^*} = \Omega K_D^{C \dashv A}$:

$$
\begin{aligned}
A^{*+} &= A^* + \left( \rho_{A^*} \left( \Theta_D(g) \frac{1}{1+(B^*/K_D^{B^* \dashv A^*})^5} + \Theta_S(g) \frac{1}{1+(B^*/K_S^{B^* \dashv A^*})^5} \frac{1}{1+(C^*/K_S^{C^* \dashv A^*})^5} \right) - \delta_A A^* \right) dt \\
&+ N_1 \sqrt{\rho_{A^*} \left( \Theta_D(g) \frac{1}{1+(B^*/K_D^{B^* \dashv A^*})^5} + \Theta_S(g) \frac{1}{1+(B^*/K_S^{B^* \dashv A^*})^5} \frac{1}{1+(C^*/K_S^{C^* \dashv A^*})^5} \right) + \delta_A A^*} \sqrt{dt}
\end{aligned}
\tag{26}
$$

A similar procedure can be followed for proteins $B$ and $C$. Therefore, multiplying the stochastic term of the Langevin equation for all proteins by $1/\sqrt{\Omega}$ is equivalent to re-scaling all variables and parameters that have the units of a protein concentration by a factor of $\Omega$. Since we set the maximal production rates and the degradation rates of all proteins to 1, the typical concentration of proteins $A$, $B$ and $C$ is normalized to 1. Re-scaling all protein concentrations and all parameters with units of protein concentration by a factor of $\Omega$ thus corresponds to setting the typical concentration of proteins to $\Omega$. In conclusion, parameter $\Omega$ of equation 23 indeed corresponds to the typical concentration of proteins.

## 4.3 Cell-to-cell coupling in the 3-gene models

A strategy that a cell can use to fight the intrinsic noise in protein concentrations is to evaluate the protein expression state of its neighbors and change its own protein expression state accordingly. In the stochastic simulations of the 3-gene models, cell-to-cell communication is modelled via a diffusion term included in the differential equations describing the dynamics of the set of protein concentrations. The higher the concentration of a given protein is in a given simulated cell, the more this protein will diffuse to neighboring simulated cells. Diffusion thus models the process of adjusting the protein concentration of a given cell according to the protein concentration of surrounding cells. The dynamics of vector $P$ in the 3-gene models is therefore given by the following differential equation:

$$
\frac{\partial P}{\partial t} = \Theta_D(g) \, D(P) + \Theta_S(g) \, S(P) - P + \eta(g, P) + D \frac{\partial^2 P}{\partial x^2}
\tag{27}
$$

where the diffusion constant $D$ controls the strength of cell-to-cell coupling. The complete stochastic simulation algorithm for the 3-gene model thus becomes:

$$P_i(x,t+dt) = P_i(x,t) + \left( \Theta_D(g(x,t)) \, D_i(P(x,t)) + \Theta_S(g(x,t)) \, S_i(P(x,t)) - P_i(x,t) + D \frac{\partial^2 P_i}{\partial x^2} \right) dt$$
$$+ \left( \frac{N_i(x,t)}{\sqrt{\Omega}} \sqrt{\Theta_D(g(x,t)) \, D_i(P(x,t)) + \Theta_S(g(x,t)) \, S_i(P(x,t)) + P_i(x,t)} \right) \sqrt{dt} \tag{28}$$

for $i = 1, 2, 3$. Note that diffusion is not included in the stochastic term, since diffusion of proteins is not a reaction in itself. In the simulations, the second spatial derivative is approximated by a second-order central finite difference with reflective boundaries.

## 4.4 Stochastic simulations of the gene-free models

Since the gene-free models simulate the dynamics of abstract variables that do not represent explicitly protein concentrations, the variance of the noise is held independent of the state of the system. The stochastic algorithm used to generate the results of *Figure 4—figure supplement 5* is therefore the following:

$$P_i(t+dt) = P_i(t) + \left( \Theta_D(g(t)) \, D_i(P(t)) + \Theta_I(g(t)) \, I_i(P(t)) + \Theta_S(g(t)) \, S_i(P(t)) - P_i(t) \right) dt$$
$$+ \frac{1}{\sqrt{\Omega}} \, N_i(t) \, \sqrt{dt} \tag{29}$$

where $i = 1, 2$, and $N(t) = [N_1(t), N_2(t)]$ is a vector containing 2 independent Gaussian random variables with mean 0 and variance 1. Parameter $\Omega$ is still included to control the level of noise, but it cannot be interpreted as the typical concentration of proteins in the system since the gene-free models do not simulate explicitly protein interactions.

## Mathematical formula for the mutual information

In deterministic simulations, the initial phase of the genetic oscillation inside a given cell determines in which part of the pattern this cell will end up. This is not necessarily the case in stochastic simulations. To quantify the robustness to noise of a given model for specific values of parameter $\Omega$ (and of the diffusion constant $D$ in the case of the 3-gene models) it is required to define a metric that measures the accuracy with which the initial phase of the genetic oscillations inside a cell predicts the region of the pattern in which this cell will end up. The specific metric used in *Figure 3*, *Figure 3—figure supplement 1*, *Figure 4—figure supplement 5* and *Figure 5—figure supplement 2* is the mutual information between the initial phase of the oscillator and the final protein expression state of the simulated cells. The mutual information $I(x,y)$ between two discrete variables $x$ and $y$ is given by the following expression:

$$I(x,y) = \sum_{y \in Y} \sum_{x \in X} p(x,y) \, \log\left( \frac{p(x,y)}{p(x) \, p(y)} \right) \tag{30}$$

where $X$ and $Y$ are the sets of possible values for $x$ and $y$, respectively. Intuitively, the mutual information between two variables quantifies the amount of information obtained on the value of the first variable by knowing the value of the second variable (and vice versa). If the logarithm is in base 2, the units of the mutual information are bits. To measure how precisely the phase of the oscillator is read to form the final pattern, variable $x$ is set to the phase of the oscillation in protein expression at the beginning of the simulation $\phi_i$, and variable $y$ is set to the protein expression state at the end of the simulation

$$I(\phi_i, P_f) = \sum_{P_f} \sum_{\phi_i} p(\phi_i, P_f) \, \log\left( \frac{p(\phi_i, P_f)}{p(\phi_i) \, p(P_f)} \right) \tag{31}$$

$$= \sum_{P_f} \sum_{\phi_i} p(\phi_i, P_f) \, \log\left( \frac{p(\phi_i, P_f)}{p(\phi_i) \, p(P_f)} \right) \tag{32}$$

$$= \sum_{P_f} \sum_{\phi_i} p(P_f|\phi_i) \, p(\phi_i) \, \log\left( \frac{p(P_f|\phi_i)}{\sum_{\phi_i} p(P_f|\phi_i) \, p(\phi_i)} \right) \tag{33}$$

To get the second equality, the fact that $p(x, y) = p(x|y)\, p(y)$ for any two variables $x$ and $y$ was used to get rid of the joint probability $p(\phi, R_i)$, which is not straightforward to evaluate directly. Similarly, the fact that $p(y) = \sum_{x \in X} p(x, y) = \sum_{x \in X} p(x|y)\, p(y)$ for any two variables $x$ and $y$ was used to get rid of $p(P_f)$, which is less easy to compute than $p(\phi_i)$. Indeed, $\phi_i$ is sampled uniformly in the simulations of the 3-gene and gene-free models, since the speed of regression of the input $g$ is constant throughout the simulations. In the 3-gene models, the different phases $\phi_i$ are defined as the different states of protein expression along the oscillation cycle generated by the dynamic module ($g = 1$). A uniform sample of $\phi_i$ is obtained by sampling this oscillation cycle at constant time intervals for a total time length of one period. In the gene-free models, the different phases $\phi_i$ are defined as the different sets of ($y$, $z$) values along the oscillation cycle generated by the dynamic module ($g = 1$). Since the oscillations are on the unit circle (centered at the origin) and have a constant speed along the cycle, sampling uniformly the angles from the positive $y$ axis (starting at 0 and stopping at $2\pi$) generates a uniform sample of $\phi_i$.

## Description of the source codes

All codes are written in the `python3` programming language (except for two `Mathematica` notebooks). Commented jupyter notebooks can be found on Github at the following address: https://github.com/laurentjutrasdube/Dual-Regime_Geometry_for_Embryonic_Patterning. This repository also contains folders with the source data files, as well as the source codes used to generate the data files.

- **3-gene_det.ipynb**
  This notebook performs deterministic simulations of the symmetric 3-gene Models 1 and 2, and deterministic simulations of the 3-gene Models 1 and 2 with Hill functions for the weights of the dynamic and static modules. It also performs a bifurcation analysis of these models using the data found in the `XPPAUTO_data` folder, which also contains the `.ode` files used to generate the data with the XPP AUTO software (*Ermentrout, 2008*). *Figure 2*, *Figure 2—figure supplements 1* and *2*, and *Figure 7—figure supplement 2* show the results obtained with this notebook.

- **3-gene_stoch.ipynb**
  This notebook performs stochastic simulations of the symmetric 3-gene Models 1 and 2, and stochastic simulations of the 3-gene Models 1 and 2 with Hill functions for the weights of the dynamic and static modules. It also generates plots of the mutual information using the data found in the `Mutual_info_data` folder, which also contains the `python` codes used to generate the data. *Figure 3* and *Figure 3—figure supplement 1* show the results obtained with this notebook.

- **3-gene_asym.ipynb**
  This notebook performs deterministic simulations of the asymmetric 3-gene Models 1 and 2. It also performs a bifurcation analysis of these models and generates plot of the mutual information using the data found in the `XPPAUTO_data` and `Mutual_info_data` folders, respectively. *Figure 5* and *Figure 5—figure supplements 1* and *2* show the results obtained with this notebook.

- **Gene-free_det.ipynb**
  This notebook performs deterministic simulations of the symmetic gene-free Models 1, 2, 3 and 4. It also performs a bifurcation analysis of these models and generates flow plots using the data found in the `XPPAUTO_data` and `Mathematica_data` folders, respectively. *Figure 4*, *Figure 4—figure supplements 1*, *2*, *3* and *4*, *Figure 4—videos 1*, *2*, *3* and *4*, and *Figure 7—figure supplement 2* show the results obtained with this notebook.

- **Gene-free_stoch.ipynb**
  This notebook performs stochastic simulations of the symmetric gene-free Models 1, 2, 3 and 4. It also generates the mutual information plots using the data found in the `Mutual_info_data` folder. *Figure 4—figure supplement 5* shows the results obtained with this notebook.

- **Gene-free_asym.ipynb**
  This notebook performs deterministic simulations of the asymmetic gene-free Models 1 and 2. It also performs a bifurcation analysis of these models using the data found in the `XPPAUTO_data` folder. Moreover, it generates plots of the flow and of the spatial wave profiles. *Figure 6*, *Figure 6—figure supplement 1*, *Figure 7* and *Figure 7—figure supplement 1* show the results obtained with this notebook.

- **Hopf_scenario_Fig1.ipynb**
  This notebook performs deterministic simulations of the gene network model evolved in silico in *François et al., 2007*. Results are shown on *Figure 1*. It also performs a bifurcation analysis of this model, shown on *Figure 1—figure supplement 1*.
- **Infinite-period_scenario_Fig1.ipynb**
  This notebook performs deterministic simulations of the infinite-period model of *Figure 1*, which is a simplified version of the model in the appendix of *Palmeirim et al., 1997*.
- **Infinite-period_scenario_Fig7.ipynb**
  This notebook performs deterministic simulations of the infinite-period model of *Figure 7*, which is adapted from *Morelli et al., 2009*.
- **Tribolium_model.ipynb**
  This notebook performs deterministic simulations of the model for Tribolium segmentation from *Zhu et al., 2017*. It also generates flow plots and computes the speed of the cells in phase space. *Figure 1—figure supplement 2* shows the results obtained with this notebook.

