## [Decision Letter]

**Acceptance summary:**

This paper applies a geometric approach to modelling the transition between dynamic gene expression and fixed cell identities, in the context of segmentation and somite formation in embryos. Such an approach is particularly useful to understand modules like the segmentation clock, which shows considerable plasticity and variation from species to species. The study highlights the possible occurrence of an interesting type of global bifurcation in such systems and indicates the kinds of experiments that would be needed to confirm or rule out the existence of this dynamical phenomenon during somite formation.

**Decision letter after peer review:**

Thank you for submitting your article "Geometric models for robust encoding of dynamical information into embryonic patterns" for consideration by *eLife*. Your article has been reviewed by two peer reviewers, and the evaluation has been overseen by a Reviewing Editor and Aleksandra Walczak as the Senior Editor. The following individual involved in review of your submission has agreed to reveal their identity: Mogens H Jensen (Reviewer #2).

The reviewers have discussed the reviews with one another, and the Reviewing Editor has drafted this decision to help you prepare a revised submission.

Both reviewers found that the manuscript had a number of interesting mathematical results and would be of considerable interest to researchers interested in nonlinear dynamical systems. However, both also agreed that the connection to biology could be improved. In your revision, please focus on the specific points in the attached reviews where the reviewers question the biological interpretation of the results, and try to provide:

1) A stronger and clearer discussion on connection of the model to biology and experiments.

2) An evaluation of how to experimentally distinguish SNIC from Hopf.

*Reviewer #1:*

The work brings together different models of the transition between dynamic gene expression and fixed cell identities, with particular significance to segmentation and somite formation, under a common framework. It builds on earlier work by the authors (Zhu et al., 2017 and Francois and Siggia, 2012) and has a detailed examination of a model which is able to show both local and global bifurcations. It has the surprising result that the global bifurcation is more generic and robust, which is explained using a geometrical approach. While the theoretical model is of interest and is covered in detail, there are only tenuous links made to the experimental literature. I think it is worth publishing with revisions but may not be suitable for *eLife* because of its strong mathematical and theoretical focus.

Substantive Concerns:

1) In both Zhu et al., 2017 (in which the model of a separate dynamic and a static module coupled through a morphogen is introduced) and Francois et al., 2007 (which evolves different networks including a repressilator + bistable system), links are made to the identities of the different components in the network and the biology (gap-genes in *Tribolium*, and *Drosophila* segmentation system). This model admittedly aims to be more general but I think it would be substantially strengthened by a clear discussion, possibly in a separate section, of the relationship of the model to the experimental literature in arthropods and possibly vertebrates.

2) Model 2 in the paper relies on a linear coupling between the two modules. A quadratic coupling leads to a Hopf bifurcation. It could be clarified how strongly the SNIC bifurcation relies on the linear coupling (both in the geometric and gene network model), and whether it is the specific form of the coupling which is important or merely its shape (e.g. will other nonlinear couplings also produce Hopf?).

3) The authors strongly argue that the SNIC scenario is more robust to noise and perturbation. However, I think the argument that the SNIC bifurcation will be generically found is weaker. As the authors point out, there is a reason why Hopf bifurcations are more generically seen. The geometrical models the authors employ to argue for the generic appearance of the SNIC bifurcation only show that the bifurcation is generic in 2D models of the kind the authors have built. They could perhaps argue that Model 2 is more likely to be the result of evolution because of its robustness in which case a comparison with Francois et al., 2007 which evolved networks and found the Hopf scenario would be useful.

4) As the authors themselves admit, the infinite period in the SNIC bifurcation is difficult to distinguish from a Hopf bifurcation. They could briefly clarify how the two bifurcations could be distinguished experimentally as done in a different context by Salvi et al., 2016.

*Reviewer #2:*

This paper deals with encoding of dynamical information into embryonic development and pattern formation. It is a very interesting paper to read and is well written. The paper relies heavily on dynamical systems theory and investigate quite a lot of bifurcations in various situations. In particular normal Hopf bifurcations (which is denoted local) versus SNIC saddle-node on an invariant circle bifurcation (which is denoted global).

I have only few critical remarks to the paper. The first is somewhat general and related to what is just stated above. The paper has a lot of interesting mathematical observations. One after the other and if one is interested in bifurcation theories of dynamical systems, it is a pleasure to read. However, I am wondering about the biological implications and how relevant they are for biological systems? Maybe some of the connections to biology are a little overstated. And therefore, one might question whether the paper is suitable for *eLife*? In *eLife* I suppose the biological side would be a key priority. In that sense the paper might fit better in a more technical journal.

One of the most beautiful examples of pattern formation in embryos is observed during somite formation. Indeed, the authors often refer to this fascinating phenomenon. The whole point about the SNIC bifurcation is that it belongs to the class of infinite period bifurcation. This is kind of obvious as the systems make a transition from a limit cycle, which gets slower and slower finally ending up in stable and unstable fixed points. And now I am a little in doubt whether this is the case for somite formation. Here, the period of oscillations indeed gets slower and slower along the mesoderm (from posterior to anterior) finally ending up where a group of cells form a somite. However, at that point the period is *not* infinite. It is longish but not infinite. Therefore, I believe that this biological example does not belong to the class of a smooth SNIC transition. Maybe it could be sub-critical or first order transition instead?

In the end the authors discuss asymmetric versus symmetric waves, which maybe again is a little bit unclear to me. Again, for the somites, the asymmetric waves appear naturally which the authors actually also state.

The authors refer to a Hopf bifurcation as local and a SNIC bifurcation as global. Indeed, that has been formulated before and the authors says, "it is associated to global changes of the flows". Sorry, I simply do not quite see what it means and would suggest just a short pedagogical explanation of this.

In conclusion, this is an interesting paper and very nicely presented with beautiful figures etc. To me it appears as an elegant presentation of several different bifurcation scenarios in various contexts. Surely of interest to people working in systems biology, mostly on the dynamical systems side. Again, I might judge it too mathematical for *eLife* but I would leave for the editors decide on this.

---

## [Author Response]

Both reviewers found that the manuscript had a number of interesting mathematical results and would be of considerable interest to researchers interested in nonlinear dynamical systems. However, both also agreed that the connection to biology could be improved. In your revision, please focus on the specific points in the attached reviews where the reviewers question the biological interpretation of the results, and try to provide:1) A stronger and clearer discussion on connection of the model to biology and experiments.2) An evaluation of how to experimentally distinguish SNIC from Hopf.

We thank the editors and the reviewers for their very insightful comments. In our revision, we address the two comments above. We summarize our modifications below

To address point 1:

– We added a new subsection “Relation to existing networks” to the Model section. We describe the *Tribolium* gap gene networks modelled with static and dynamic modules that first inspired this work. We also explain why the *Notch* signalling pathway is an ideal candidate for a single family of genes regulating both the static and dynamic regimes of vertebrate segmentation.

– We also extensively rewrote the Discussion to better connect to biology (in relation to point 2, see below). We extensively describe features of metazoan segmentation (including mutants) suggesting the existence of SNIC bifurcations. We also discuss evo-devo implications, connecting the robustness of the SNIC-based models to the experimentally observed variability and plasticity of segmentation mechanisms. We believe those new discussions of data (including *Wnt* mutants) connect evo-devo to dynamical system theory in a highly original way and are thus of interest for a biological audience.

To address point 2:

– We rewrote our last section in Results to focus more explicitly on the “Hopf vs. SNIC” question. We discuss and provide metrics to experimentally distinguish segmentation mechanisms based on Hopf and SNIC bifurcations. Specifically, we explain the difference in the dynamics of the shape of the oscillations that distinguish the two bifurcations. We added a new supplementary figure showing how we can visualize this difference in both the 3-gene and the gene-free models. Furthermore, we describe how the period divergence occurring during SNIC bifurcations makes the local spatial wavelength of oscillations reach the somite length in a continuous fashion. Therefore, comparing the local wavelength of propagating waves observed during vertebrate segmentation to the wavelength of anterior markers in the pattern might help distinguish finite-period scenarios from infinite-period scenarios. Finally, we discuss how changes of the shape of spatial wave profiles characteristic of patterning mechanisms based on Hopf and SNIC bifurcations might be observable in experiments.

– We also added a new subsection “Experimental evidence” in the Discussion that reviews the current evidence associated to each of the three metrics summarized above. First, we argue that the gradual reinforcement and refinement of the in situ oscillation patterns of the Notch signalling pathway observed in virtually all model organisms is more compatible with a SNIC. Second, we revisit experimental papers in which we believe that evidence for period divergence can be found, notably on *Mesp2* patterns in both wild-type and beta-catenin mutants in mice. Third, we argue that there is effectively an asymmetry in the spatial wave profiles during somite formation, a possible readout being the downstream pattern of *Cadherin*.

Reviewer #1:[…]Substantive concerns:1) In both Zhu et al., 2017 (in which the model of a separate dynamic and a static module coupled through a morphogen is introduced) and Francois et al., 2007 (which evolves different networks including a repressilator + bistable system), links are made to the identities of the different components in the network and the biology (gap-genes in Tribolium, and *Drosophila* segmentation system). This model admittedly aims to be more general, but I think it would be substantially strengthened by a clear discussion, possibly in a separate section, of the relationship of the model to the experimental literature in arthropods and possibly vertebrates.

As suggested by the reviewer, we added a new subsection in the “Model” section in which we discuss how our model relates to components of existing gene networks. We have more explicitly detailed the *Tribolium* system, and we now explain in detail how our approach could apply to the Notch signalling pathways in somitogenesis. We also expanded the Discussion to point out other possible links with biology, as summarized in the response to the main concerns listed above.

2) Model 2 in the paper relies on a linear coupling between the two modules. A quadratic coupling leads to a Hopf bifurcation. It could be clarified how strongly the SNIC bifurcation relies on the linear coupling (both in the geometric and gene network model), and whether it is the specific form of the coupling which is important or merely its shape (e.g. will other nonlinear couplings also produce Hopf?).

Intuitively, Hopf bifurcation arises when the weights of the dynamic and static modules become small compared to the weight of the rest of the dynamics (i.e. the other parts of the ODEs describing the dynamics of the system such as degradation rates). So the specific form does not matter, rather the shape matters. Indeed, other non-linear coupling can create similar effects. For instance, with Hill functions, we get both SNIC and Hopf bifurcations with the same shape for the two weights. The Hopf is obtained by shifting the weight of the static term towards smaller values of the control parameter, effectively generating the required intermediate regime.

The gene-free model helps clarifying these concepts, and we have expanded the explanation. To get a Hopf, we need three things. First, we need to introduce a 3^rd^ module (that we call the “intermediate module”) characterized by a single fixed point. Without this module, we consistently get a SNIC bifurcation for all forms of coupling that we tested, including linear and non-linear couplings. Second, we need to set the weight of the intermediate module to 0 for g=1 and g=0, so that we get the oscillations of the dynamic module at the beginning of the simulation and the bistability of the static module at the end of the simulation. We achieve this by using a weight of 2^nd^ order in g for the intermediate term: g(1-g). Third, we need to make the weights of the dynamic and static modules smaller that the weight of the intermediate module for values of g around 0.5, which is achieved by using cubic weights. Weights of higher order in g also lead to a Hopf bifurcation. Interestingly, linear weights lead to a SNIC bifurcation. We added a supplemental figure that illustrates these results, Figure 4—figure supplement 1, with the accompanying description in the main text.

3) The authors strongly argue that the SNIC scenario is more robust to noise and perturbation. However, I think the argument that the SNIC bifurcation will be generically found is weaker. As the authors point out, there is a reason why Hopf bifurcations are more generically seen. The geometrical models the authors employ to argue for the generic appearance of the SNIC bifurcation only show that the bifurcation is generic in 2D models of the kind the authors have built. They could perhaps argue that Model 2 is more likely to be the result of evolution because of its robustness in which case a comparison with Francois et al., 2007 which evolved networks and found the Hopf scenario would be useful.

We agree with the reviewer that there is an interesting evolutionary question related to our work. One possible reason why Francois et al., 2007, did not find the SNIC model is that the transcriptional model used in that paper might be too simple. Indeed, there was only one enhancer per gene in that paper. In silico evolution in this context could not find a way to easily interpolate between two (non-existing) enhancers, and therefore had to engineer incrementally both attractors and a “simple” transition between them (via Hopf/saddle node bifurcations). This is a good model to build segmentation “from scratch”, possibly for an ancestral form (an “ursegmentation” process). Also if the gradient of g is steep enough, there is no difference between SNIC and Hopf, which could be a reason why SNIC bifurcations would not spontaneously evolve. Adding more sophisticated steps to the evolution algorithm, like the possibility to have multiple enhancers controlling the same family of genes, might allow for more combinatorial use of dynamical attractors, which in turn might lead to SNIC bifurcations. We believe a complete study of such evolutionary effects is beyond the scope of our current manuscript, but our plan is to upgrade the current version of the evolutionary algorithm used in Francois et al., 2007, to (1) revisit those ideas by using a richer evolutionary grammar that allows combinations of enhancers, and (2) by using more variable shapes of input gradients, to explicitly select for robustness. At this stage, we have added a new subsection titled “Evolution and developmental plasticity” in the Discussion, in which we describe this possible explanation of the apparent discrepancy between the findings in François et al., 2007, and our present findings, and link our results to the important biological notion of developmental plasticity.

4) As the authors themselves admit, the infinite period in the SNIC bifurcation is difficult to distinguish from a Hopf bifurcation. They could briefly clarify how the two bifurcations could be distinguished experimentally as done in a different context by Salvi et al., 2016.

We have expanded the Results and the Discussion to address this comment. We thank the reviewer for pointing out the Salvi et al., 2016, reference, that we now cite and use. Unfortunately, the methods proposed in Salvi et al., 2016, cannot be directly implemented in the context of embryonic development because there is no obvious way to experimentally modify the value of the control parameter, while they can control the load stiffness and the external force applied on the hair bundle. So we address this comment by using the position in PSM as a proxy for this control parameter.

One metric used in Salvi et al., 2016, is the evolution of the distribution of points on the limit cycle as a function of the control parameter. We can visualize this in our models (and possibly experimentally) and added a new supplemental figure: Figure 7—figure supplement 2 to illustrate this. We conclude that the experimentally observed distribution is much more consistent with a SNIC: it is a well-known fact that amplitude of the oscillations increase before stabilization, which would correspond to a stretched distribution seen for SNIC but not for Hopf.

Another metric used in Salvi et al., 2016, to identify the type of bifurcation is the evolution of the frequency, which presents a discontinuity for Hopf bifurcations, but continuously goes to 0 for SNIC bifurcations. Giudicelli et al., 2007, proposed a simple model to infer the period of the clock from the local spatial wavelength, and we now discuss extensively how spatial wavelength connect to period divergence. In the discussion, we explain in detail how it can be observed experimentally, e.g. via gene *Mesp2*, and further explain how *Wnt* mutants (both in insects and vertebrates) are suggestive of such period divergence.

SNIC bifurcations are also accompanied by characteristic changes of the shape of spatial wave profiles that might be observable in experiments. We now describe more clearly these asymmetries in the main text and in the schematic of Figure 7. See also our reply to point 3 of reviewer 2, who asked specifically to clarify what we mean by “wave asymmetry”. We also relate this asymmetry in the Discussion to actual biological outputs such as *Cadherin* patterns.

Reviewer #2:[…]I have only few critical remarks to the paper. The first is somewhat general and related to what is just stated above. The paper has a lot of interesting mathematical observations. One after the other and if one is interested in bifurcation theories of dynamical systems, it is a pleasure to read. However, I am wondering about the biological implications and how relevant they are for biological systems? Maybe some of the connections to biology are a little overstated. And therefore, one might question whether the paper is suitable for eLife? In eLife I suppose the biological side would be a key priority. In that sense the paper might fit better in a more technical journal.

We have now added many biological details, considerably expanding the Results and the Discussion to address those concerns. See our response to reviewer 1 points 1 and 4.

One of the most beautiful examples of pattern formation in embryos is observed during somite formation. Indeed, the authors often refer to this fascinating phenomenon. The whole point about the SNIC bifurcation is that it belongs to the class of infinite period bifurcation. This is kind of obvious as the system makes a transition from a limit cycle, which gets slower and slower finally ending up in stable and unstable fixed points. And now I am a little in doubt whether this is the case for somite formation. Here, the period of oscillations indeed gets slower and slower along the mesoderm (from posterior to anterior) finally ending up where a group of cells form a somite. However, at that point the period is not infinite. It is longish but not infinite. Therefore, I believe that this biological example does not belong to the class of a smooth SNIC transition. Maybe it could be sub-critical or first order transition instead?

This is a very interesting point, and we now extensively revisit the experimental literature to address it. The slowdown of oscillations appears quite variable between species. In the subsection “Experimental evidence” added to the Discussion, we argue that the mouse segmentation clock seems (at first) more consistent with the Hopf scenario, while other species, including the zebrafish and the snake, have several propagating waves within the PSM. We more explicitly relate now wavelength to period and recall that it was argued by other groups that the period was indeed becoming infinite for zebrafish. We also argue that the *Mesp2* expression pattern in the anterior PSM could serve as a read-out of this spatial wavelength and discuss evidence for infinite period. We also now notice that the accompanying increase in amplitude is more consistent with SNIC.

To further address the reviewer’s comments, we added two new supplemental figures where we show the results of simulations of two geometric models that display subcritical Hopf bifurcations. To do this, we have to change the dynamical module to make it more non-linear. On Figure 4—figure supplement 3, we show that indeed we can get a subcritical Hopf bifurcation (followed by a saddle-node of limit cycles) if we include the intermediate module and use cubic weights for the dynamic and static modules. However, there is a wide range of values of the control parameter for which only one fixed point is present, as for the supercritical Hopf bifurcations. This makes the pattern very fragile to noise, as demonstrated by our mutual information metric shown on Figure 4—figure supplement 5. On Figure 4—figure supplement 4, we show that with this different dynamic module, we can even get two simultaneous subcritical Hopf bifurcations, followed by two saddle node of limit cycles bifurcation if we use linear weights for the dynamic and static modules and set the weight of the intermediate module to 0. This is the strategy we followed to generate a SNIC bifurcation. This behaviour essentially is the “subcritical” equivalent of the SNIC bifurcation, and as such is also robust to noise. However, we notice that such bifurcations (like l subcritical bifurcations) are more “complex” than simpler cases (in the sense they have more complex normal forms), and while those bifurcations are possible in our framework, we focus in the main text on the simpler cases.

Lastly, we are not entirely sure how “1^st^ order vs. 2^nd^ order transition” framework can be applied to our models, since they describe the dynamics of gene concentrations (or geometric variables that serve as proxies for gene concentrations), and not physical variables that could be linked to states of matter and/or to a latent heat.

In the end the authors discuss asymmetric versus symmetric waves, which maybe again is a little bit unclear to me. Again, for the somites, the asymmetric waves appear naturally which the authors actually also state.

We have addressed this comment by explaining in more detail the shape of the asymmetric wave profiles, as well as where these asymmetric waves come from. We also modified the schematic of asymmetric wave profiles of Figure 7 to make it more explicit. We now use a Van der Pol model that provides a better visualization of the asymmetry in the wave profile. We also explain that those spatial asymmetries are likely due to the asymmetries of the limit cycles of relaxation oscillators, where a system jumps between two (or more) states.

The authors refer to a Hopf bifurcation as local and a SNIC bifurcation as global. Indeed, that has been formulated before and the authors says, "it is associated to global changes of the flows". Sorry, I simply do not quite see what it means and would suggest just a short pedagogical explanation of this.

We have reformulated this passage to make the explanation of the difference between local and global bifurcations more pedagogical.

In conclusion, this is an interesting paper and very nicely presented with beautiful figures etc. To me it appears as an elegant presentation of several different bifurcation scenarios in various contexts. Surely of interest to people working in systems biology, mostly on the dynamical systems side. Again, I might judge it too mathematical for eLife but I would leave for the editors decide on this.

We hope that our extensive revision, including new comments on the existing biological literature and methods to generate and study the bifurcations, will be of interest for a general audience of both theorists and (developmental) biologists, justifying publication in *eLife*.